# Marrying Generative Model of Healthcare Events with Digital Twin of Social Determinants of Health for Disease Reasoning

Ziquan Wei [1]  Tingting Dan [2]  Guorong Wu [2] [1]

## Abstract

Despite the central role of sensor-derived measurements such as imaging traits and plasma biomarkers in biomedical research and clinical practice, existing generative models for disease prediction largely depend on event-level representations from hospital and registry data. Given the multifactorial nature of human disease, the absence of explicit modeling of social determinants of health (SDoH) limits the capacity for personalized disease modeling and clinical decision support. To address this limitation, we propose a generative model with ICD-coded proxies of SDoH for *in silico* modeling of disease reasoning, a conditioned latent diffusion framework that establishes the connection between multi-organ sensor data with tokenized healthcare events. Specifically, we introduce a novel geometric diffusion model to characterize the temporal evolution of complex data representation such as brain networks (region-to-region connectivity encoded in a graph), in parallel with diffusion models for tabular data from other organ systems. Together, we integrate the generative model with digitalized SDoH proxies (coined *DiffDT*) for simulated intervention and reasoning of future disease trajectories. We conduct extensive experiments on the UK Biobank (UKB) dataset, which contains organ-specific imaging traits, including brain (44,834), heart (23,987), liver (28,722), and kidney (32,155), along with nearly 500k medical history sequences (age range: 25~89 years). Our *DiffDT* achieves significant improvements over state-of-the-art human disease autoregressive models and imaging trait generative baselines.

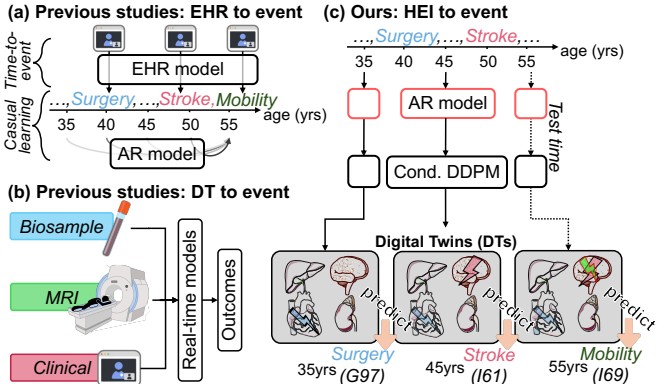

*Figure 1.* The flowchart of disease prediction in previous studies by (a) EHR-to-event methods and (b) digital twin (DT)-to-event methods, respectively, and (c) ours using ICD-coded social determinants of health (SDoH) proxies-to-event method (abbreviated SDoH-to-event throughout). Instead of predicting next disease solely derived from (a) the EHR history or (b) the combination of digitized *in-vivo* biomarkers, we introduce (c) the DTs conditioned on SDoH proxies into the AR model so that it predicts through sensor-derived physiological state encoded by the human DT. The disease trajectory, which leads to Mortality, can therefore be connected by the generative multi-organ DTs in the context of past diseases denoted by cyan and purple.

## 1. Introduction

Artificial intelligence shows strong potential to aid healthcare decision making by learning patterns of disease progression from large cohorts of electronic health records (EHRs), such as hospital inpatient data, primary care records, and registry data (Shmatko et al., 2025). Most existing approaches focus on modeling relationships between medical events and healthcare outcomes derived from EHR data. Recent advances have demonstrated the effectiveness of human disease autoregressive (AR) models (Steinberg et al., 2024; Shmatko et al., 2025), which formulate disease progression as a next-token prediction problem using Transformer architectures (Vaswani et al., 2017). These models capitalize on large-scale longitudinal EHRs to learn time-to-event correspondence or the causality between diseases as shown in Fig. 1 (a), encompassing over seven million temporally ordered clinical events encoded using the International Classification of Diseases (ICD) taxonomy. Although lifestyle factors and co-morbidity risks are often taken into account, the event-

[1]Department of Computer Science, UNC Chapel Hill, USA [2]Department of Psychiatry, UNC Chapel Hill, USA. Correspondence to: Guorong Wu <grwu@med.unc.edu>.

*Proceedings of the 43rd International Conference on Machine Learning*, Seoul, South Korea. PMLR 306, 2026. Copyright 2026 by the author(s).

driven learning framework lacks direct measurements of underlying physiological processes to calibrate uncertainties. As a result, current methods are largely trained to model healthcare utilization patterns rather than an in-depth understanding of disease etiology, which is responsible for the limited power in personalized long-horizon disease forecasting and clinically valid associative modeling of biological pathways.

In contrast, biomarkers provide a direct window into underlying physiological and pathological processes and are therefore essential for accurate disease modeling and prediction. For example, the advance of *in-vivo* medical imaging technologies such as MRI and CT provides spatially resolved measures of structural and functional alterations across interconnected organ systems. In this context, tremendous efforts have been invested in the area of digital twin (DT) that aim to digitize diverse biomarker readouts and model the intervention effects as demonstrated in Fig. 1 (b), with broad applications in precision medicine (Sahal et al., 2022; Venkatesh et al., 2022), clinical trials (Kolla et al., 2021; Qi & Cao, 2023), biomanufacturing (Park et al., 2021), organ simulation (Le et al., 2021), and socio-ethical benefits/risk assessment (Popa et al., 2021). However, limited attention has been paid to empowering digital twins with the ability to infer disease progression across the lifespan. As a result, current DT approaches lack the ability to model the longitudinal causal chain where past medical history might shape the current state of physiological condition, which in turn acts as the biological mediator for future disease onset.

Taken together, we present an integrated learning framework to combine the power of event-driven AR model and *in silico* modeling of ICD-coded proxies of social determinants of health (SDoH). The model overview is shown in Fig. 1 (c), which is characterized by two components: (1) an adaptive tokenization method for the embedding of medical history coded with ICD, denoted by AR model and (2) a conditional latent diffusion model $\epsilon_\theta(\text{DT}, t, \text{ICD})$, denoted by 'Cond. DDPM', which characterizes how a medical event occurring at time $t$ influences subsequent disease progression through sensor-derived physiological state encoded by the human digital twin.

Since most diseases arise from system-level interactions rather than isolated organ dysfunction, the sensor data comprise biomarker readouts from multiple organs, including the brain, heart, liver, and kidney, represented across diverse data modalities and structures. We deploy the conventional diffusion-based generative model, such as DDPM (Ho et al., 2020), to learn the distribution of tabular biomarker data such as myocardial wall thickness and liver iron concentration. Furthermore, we devise a geometric generative model for topological data, such as a functional brain network, which is essentially a covariance matrix recording func-

tional co-activations across brain regions. To that end, we cast each brain functional connectivity as a data instance on the Riemannian manifold of symmetric positive-definite (SPD) matrices, yielding a geometric diffusion model.

By capitalizing on rich data collection in UK Biobank (UKB), we train our diffusion-based DT (coined *DiffDT*) on EHR data and multi-organ biomarkers. Performance on disease prediction and reasoning has been evaluated on over 1,000 diseases compared with autoregression models and foundation models. The **primary technical contribution** of this work is an SPD-VQVAE built on Cholesky decomposition that maps brain functional connectivity onto a Riemannian manifold-aware latent space, enabling a Cholesky LDM that respects SPD geometry while remaining computationally tractable. Unlike prior AR-diffusion hybrids designed for Euclidean data, our SPD-VQVAE is purpose-built for the non-Euclidean geometry of brain networks, which is the central technical bridge between discrete clinical events and continuous physiological manifolds. Additional **technical contributions** include (1) an adaptive tokenization method for the embedding of medical history; (2) the Cholesky LDM with SPD-VQVAE-Dual decoder that, together with Theorem 3.1, guarantees generated digital twins reside on the SPD manifold without incurring $O(N^3)$ cost; and (3) a probabilistic mediation inference mechanism in *DiffDT* that enables multi-pathway disease reasoning in the AR paradigm.

## 2. Relevant works and their limitations

**Generative models based on EHRs.** ICD is the global standard for health data, clinical documentation, and statistical aggregation. It serves as a universal language that translates complex medical diagnoses into alphanumeric codes, allowing the EHR written in various natural languages to be standardized across different medical systems. The $10^{th}$ revision is used here (Hirsch et al., 2016). This scientifically rigorous system, maintained by the World Health Organization (WHO), adheres to an alphanumeric hierarchical structure, which provides categories of more than 155,000 diseases with granularity and specificity (Jetté et al., 2010), as well as the possibility of the tokenization of arbitrary diseases for the training of AR models (Steinberg et al., 2024; Shmatko et al., 2025). MOTOR (Steinberg et al., 2024) and Delphi (Shmatko et al., 2025), based on perspectives of time-to-event pretraining and probabilistic mediation, respectively, have demonstrated the potential of applying generative models to the prediction of human disease timelines. However, these studies ignored the semantic difference between events, which leads to multi-pathway probabilistic mediation between human diseases. The attempt of learning probability $P(\text{Future Disease}|\text{Past History})$ directly is challenging, due to the lack of conditional proba-

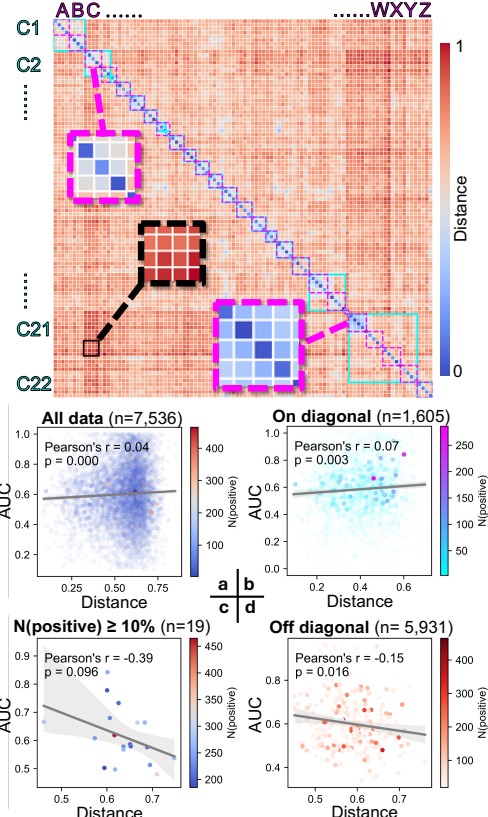

*Figure 2.* The causality between human diseases is correlated with the AR model performance. **Top**: The semantic adjacency matrix between human diseases is represented by the normalized distance between text embeddings of the meaning, where ICD chapters from I (C1) to XXII (C22) and ICD 1st level groups from A to Z are marked by cyan boxes and magenta boxes, respectively. **Bottom**: The accuracy of a pretrained AR model (Shmatko et al., 2025) predicting the next token is dependent on the semantic distance to the previous token, where each dot is an evaluation for a pair of ICD events and $n$ indicates the dot number of the scatter plot.

bility $P(\text{Biomarker}|\text{Past History})$ that drives the actual biological mechanism of disease progression.

**AR-diffusion hybrids and digital twin literature.** Hybrid AR-diffusion architectures have emerged in vision-language settings, e.g., real-time streaming sign language production (Ye et al., 2025) and the unified vision-language-action model HybridVLA (Liu et al., 2025), which couple autoregressive and diffusion generators in Euclidean spaces. Our setting differs in two essential respects: (i) the data manifold is non-Euclidean, since brain functional connectivity lives on the SPD cone, so standard noise injection corrupts the geometric structure; and (ii) the conditioning side is a long-horizon, sparsely sampled clinical event stream rather than text or vision tokens. *DiffDT* addresses both by training the AR backbone over irregular ICD timelines and routing diffusion through the SPD-VQVAE latent so the resulting digital twins are guaranteed to reside on the SPD manifold. Compared to prior digital twin work focused on subject-

specific physical simulation or pretrained representation extraction, *DiffDT* is the first framework, to our knowledge, that bridges discrete EHR histories with multi-organ continuous biomarker manifolds for longitudinal disease reasoning.

**Pitfall of current generative models for disease prediction.** Since current methods are predominantly trained on EHR data, the statistical power of EHR-drived tokens across human diseases becomes the key for accurate disease prediction. The semantic relationship between diseases is preliminarily demonstrated by the distance between ICD representations in the text space to reflect both direct and multi-pathway probabilistic mediation between human diseases. As shown in Fig. 2 top, we demonstrate the semantic distance between text embeddings of the description of top-level diagnostic codes of ICD-10 using pretrained Qwen3 (Yang et al., 2025). There are $n =2,066$ ICD top-level codes, and the matrix is linearly downsampled to $100 \times 100$ for visualization. The direct semantic relation represented by small distances (blue-ish cells) is consistent with the predefined clinical categories (cyan and magenta boxes in the distance matrix), confirming the granularity of the ICD standard.

Following this clue, evidence of the performance degradation by current generative models based on ICD only (Shmatko et al., 2025) is shown in the distance vs. area under the ROC Curve (AUC) plots of Figure 2. Using the evaluation method in (Shmatko et al., 2025), the accuracy of disease prediction is evaluated by AUC score for each ICD-coded disease. Although the semantic adjacency matrix contains the distance between more than four million pairs of ICD codes, there are $n =7,536$ pairs of ICD codes having enough sample size of positive (i.e., diseased) subjects to compute the AUC score. Therefore, referring to adjacency matrix of human diseases in Fig. 2 top, we explore the relationship between semantic distances and AUC scores within four interesting groups of ICD pairs: (1) All $n =7,536$ ICD pairs using all data (Fig. 2a); (2) Past and future diseases are from the same ICD Chapter (Fig. 2b), i.e., ICD pair within the cyan boxes on the diagonal line; (3) ICD pairs with more than 10% subjects diagnosed with the new disease (Fig. 2c); (4) Past and future diseases fall into different ICD chapters (Fig. 2d), which is off-diagnal pairs. It is clear that while AUC-distance is not substantially correlated among all ICD pairs or on the diagonal, they manifest significant negative correlations for those ICD pairs with enough positive samples or off the diagonal, meaning the model is better at predicting semantically close diseases than distant ones with multi-pathway probabilistic mediation. This observation highlights the need for methods addressing multi-pathway human disease reasoning, potentially by leveraging more reliable modalities such as biomarkers.

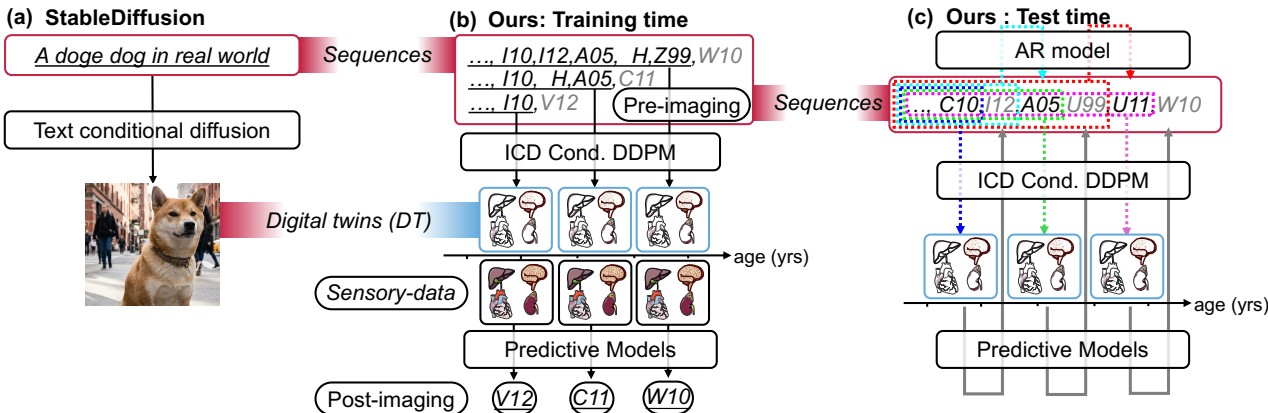

*Figure 3.* The framework of *DiffDT* for human disease reasoning by cooperating generative and AR models. (a) The StableDiffusion method is achieved by paired data of text sequences and real-world images during training. (b) *DiffDT* is feasible since there are cross-sectional paired data of human disease history and organ images for training generative models, i.e., conditional (cond.) DDPM, as well as training the models to predict the next event from the sensory biomarkers. (c) During inference, the generative and predictive models cooperate with the AR model for reasoning multi-pathway causality between past (black ICD) and future (gray ICD).

## 3. Methods

**Problem definition.** We consider a set of $N$ subjects, their medical records $\mathbf{S}$, and biomarker data $\mathbf{B}$. $\mathbf{S}^{(i)} \in \mathbb{R}^T$ denotes subject's records (subscript $i$ denotes subject index) that consists of a sequence of event indices, where $T$ denotes the length of ICD sequence. Each $\mathbf{S}^{(i)}$ is defined by ICD 10-th revision, with the timestamp sequence $\tau^{(i)} \in \mathbb{R}^T$ in years. Biomarker $\mathbf{B}^{(i)} = \{\mathbf{\Gamma}^{(i)}, \mathbf{M}^{(i)}\}$ consists of tabular readout $\mathbf{\Gamma}^{(i)} \in \mathbb{R}^{d_{organ}}$ and topological data $\mathbf{M}^{(i)} \in \mathbb{R}^{N \times N}$ denoting brain functional connectivity derived from functional neuroimaging data[1], where $d_{organ}$ and $N$ denote the number of traits and brain regions, respectively.

Unlike existing work (Steinberg et al., 2024) only focusing on predicting preselected diseases, we seek to cover human disease reasoning that involves all coded diseases in ICD to complete the probabilistic mediation inference: $s_t^{(i)} = \phi(\mathbf{S}_{<t}^{(i)})$, where $t \leq T$ and $\phi$ is the learnable AR model. Assume $\mathbf{M}^{(i)}$ is captured at age $\tau_t^{(i)}$ and paired with $\mathbf{S}^{(i)}$, our model *DiffDT* has three major learning components: **(1)** Tokenizing each index in $\mathbf{S}^{(i)}$ as an event embedding containing the causality information between their past events. **(2)** Generating the latent features of $\mathbf{B}^{(i)}$, namely DT, given $\mathbf{S}_{<t}^{(i)}$, where the multi-pathway causality occurs at an age $t$. **(3)** Predicting next ICD event $s_{t+1}^{(i)}$ from generated $\mathbf{B}_t^{(i)}$.

### 3.1. Framework

As shown in Fig. 3, the core innovation of our framework is the cooperation of the generative and AR models at

---

[1]Brain functional connectivity is an $N \times N$ covariance matrix where each element in $\mathbf{M}^{(i)}$ measures the correlation of neural activity signals between distinct brain regions.

ages where multi-pathway causality is present. Unlike standard autoregressive models that predict $P(\text{Future}|\text{History})$ directly via pure causality, our approach models the biological mediation inference as $P(\text{Future}|\text{Biomarker}) \cdot P(\text{Biomarker}|\text{History})$. As shown in Fig. 3(a) and (b), as the text sequences are paired with the corresponding images for vision-language applications, disease trajectories are also paired with real-world sensory data, i.e., multi-organ biomarkers.

**Multi-organ DT generation.** For a subject at age $t$, we use the conditional diffusion models (see Section 3.3) to generate synthetic digital twins $\mathbf{M}_t$ and $\mathbf{\Gamma}_t$ given their medical history $\mathbf{S}_{<t}$. These generated organs act as the biological phantom of the cumulative risk factors. As shown in Fig. 3 (b), diffusion models are trained only on the age at which each subject has the ground truth of biomarkers.

**Predictive model finetuning.** To implement the second term of the pathway, $P(\text{Future}|\text{Biomarker})$, we leverage pretrained organ-specific Foundation Models (FM). As shown in Fig. 3 (b), we finetune FMs on a downstream classification task, predicting the next ICD token $s_{t+1}$ given the true observed brain functional connectivity matrix $\mathbf{M}_t$.

**Inference.** During testing shown in Fig. 3 (c), the pipeline operates sequentially for every multi-pathway disease reasoning that is marked in gray in the ICD sequence. First, the AR model encodes the past ICD sequence produce medical history embeddings (see cyan and red boxes). Then, the diffusion models generate the hypothetical states of multi-organ DTs for the next time step (see blue, green, and purple boxes). Finally, the finetuned FM analyzes this generated DT to predict the onset of future diseases, denoted by the gray arrow.

## 3.2. Adaptive Medical History Tokenizer and AR model

Standard EHR data consists of sparse events, where the time intervals between diagnoses vary significantly. To effectively condition the diffusion model on this irregular timeline, we propose an adaptive tokenization strategy that standardizes the temporal resolution. Then, an AR model $\phi$ can be pretrained by the supervision of next-token prediction to produce event embeddings $\hat{\mathbf{y}} \in \mathbb{R}^{T \times d_\phi}$, where $d_\phi$ denotes the hidden dimensionality of $\phi$.

We construct a vocabulary $\mathcal{V}$ consisting of the unique top-level ICD codes plus a code for healthy. We define a unified temporal grid spanning the age range of the cohort,

$$\tau = (\tau_t \mid (\tau_{t+1} - \tau_t) \in \{0, 1\}, \forall \tau_{\min} \le t \le \tau_{\min}), \quad (1)$$

so that $\tau$ contains every integer within the age range with simultaneous events allowed. If a subject $i$ records a disease code $c \in \mathcal{V}$ at age $\tau_t$, we set $s_t^{(i)} = c$, otherwise, we assign a special token $s_t^{(i)} = $ healthy.

We pretrain a Transformer encoder, $\phi : \mathbf{y}_t \mapsto \text{logits}_{t+1} = \text{MLP}(\hat{\mathbf{y}})$, with multihead causal self-attention as an AR model on sequences from over 500,000 subjects to learn a causality-aware embedding space, where the input $\mathbf{y}_t = \text{Embed}_{\text{ICD}}(s_t) + \text{Embed}_{\text{age}}(\tau_t)$, a token embedding plus a time embedding in the space of $\mathbb{R}^{d_\phi}$. The model optimizes the next-token prediction objective:

$$\mathcal{L}_{\text{AR}} = -\sum_t \log p(s_{t+1}|s_0, \ldots, s_t; \phi). \quad (2)$$

## 3.3. Digital Twin for ICD-Coded SDoH Proxies

### 3.3.1. DIFFUSION MODELS FOR TABULAR MULTI-ORGAN DATA

**Forward Process (Noise Injection).** We follow the idea from (Shi et al., 2024) to allow sequence-conditioned tabular data generation. Given $\mathbf{z}_t^{tab} = [\mathbf{\Gamma}_t^{(i)}, \mathbf{S}_t^{(i))}]$ and $\mathbf{z}_0^{tab} = [\mathbf{\Gamma}^{(i)}, \mathbf{S}^{(i))}]$, we utilize a hybrid continuous-time diffusion framework that applies distinct stochastic processes to numerical and categorical modalities simultaneously. For continuous variables ($\mathbf{\Gamma}$), the forward process is modeled as a Stochastic Differential Equation (SDE) using the Variance Exploding (VE) formulation (Song et al., 2020). The system gradually injects Gaussian noise according to a learnable noise schedule $\sigma^{num}(t)$, where $num$ stands for numerical. For discrete variables ($\mathbf{S}$), the forward process is defined as a discrete state-space diffusion modeled as an absorbing process. Rather than adding Gaussian noise, the process smoothly interpolates between the data distribution and a target distribution where all probability mass is assigned to a special token $A$:

$$\mathbf{\Gamma}_t = \mathbf{\Gamma} + \sigma^{num}(t)\epsilon, \quad \text{where } \epsilon \sim \mathcal{N}(0, I)$$
$$q(\mathbf{S}_t|\mathbf{S}) = \text{Cat}(\mathbf{S}_t; \alpha_t \mathbf{S} + (1 - \alpha_t)A) \quad (3)$$

where $\sigma(t)$ is the learnable power-mean noise schedule, $A = $ [MASK] which denotes an extra class for masked columns, $\alpha_t$ is the log-linear masking schedule, and $\text{Cat}(\cdot)$ is the outcome distribution: The original token $\mathbf{S}$ with probability $\alpha_t$, and the token $A$ with probability $1 - \alpha_t$.

**Reverse Process (Denoising).** The reverse process is a generative operation in which Transformer models (Vaswani et al., 2017) approximate the true posterior to recover $\mathbf{z}_0^{tab}$ from the noisy state $\mathbf{z}_t^{tab}$ for $\mathbf{\Gamma}_t$ and $\mathbf{S}_t$, respectively.

*Numerical Denoising for $\mathbf{\Gamma}_t$:* The model approximates the score function, gradient of the log-density, to solve the probability flow Ordinary Differential Equation (ODE), i.e., this reverse processing. The Transformer model $\mu_\theta^{num}(\mathbf{\Gamma}_t, t)$ predicts the noise component $\epsilon$ to denoise the tabular data.

*Categorical Unmasking for $\mathbf{S}_t$:* The Transformer model $\mu_\theta^{cat}(\mathbf{S}_t, t)$ estimates the clean data $\mathbf{S}$ (logits) directly, where '$cat$' stands for categorical. The reverse transition probabilities $p_\theta(\mathbf{S}_s|\mathbf{S}_t)$ are computed using the posterior form, which dictates that if a token is unmasked, it remains fixed. Otherwise, there is a probability derived from $\alpha_t$ to unmask it to the predicted token.

**Training Objective.** The objective for numerical features is a simple Mean Squared Error (MSE) loss, minimizing the difference between the predicted noise and the injected noise $\epsilon$. This is equivalent to score matching.

$$\mathcal{L}_{num} = \mathbb{E}||\mu_\theta^{num}(\mathbf{\Gamma}_t, t) - \epsilon||_2^2 \quad (4)$$

The objective for categorical features optimizes the Evidence Lower Bound (ELBO) in the continuous-time limit. It maximizes the log-likelihood of the true class $x_0$ predicted by the model.

$$\mathcal{L}_{cat} = \mathbb{E} \sum_t \frac{\alpha_t'}{1 - \alpha_t} \log\langle\mu_\theta^{cat}(\mathbf{S}_t, t), \mathbf{S}\rangle \quad (5)$$

Last, the training objective for multi-organ data generation combines losses for both data types, $\mathcal{L}_{tab} = \lambda_{num}\mathcal{L}_{num} + \lambda_{cat}\mathcal{L}_{cat}$, weighted by $\lambda_{num}$ and $\lambda_{cat}$.

**Tabular Biomarkers Generation.** The generation of $\mathbf{\Gamma}$ is defined as a classifier-free table imputation task with the conditions of observed ICD sequences $\mathbf{S}$. In the reverse diffusion process, this is achieved by conducting denoising sampling for the organ-specific columns while the observed components $\mathbf{S}$ are fixed. The Classifier-Free Guidance (CFG) (Shi et al., 2024) is adapted here. Instead of training a separate classifier $p(\mathbf{S}|\mathbf{\Gamma})$, it modifies the sampling score by combining a conditional estimate and an unconditional estimate. The guided conditional distribution $\tilde{p}(\mathbf{\Gamma}_t|\mathbf{S})$ is derived via Bayes' rule and parameterized by a guidance strength $\omega > 0$

$$\log \tilde{p}(\mathbf{\Gamma}_t|\mathbf{S}) = (1 + \omega)\log p_\theta(\mathbf{\Gamma}_t|\mathbf{S}) - \omega \log p_\phi(\mathbf{\Gamma}_t) \quad (6)$$

where $p_\theta(\mathbf{\Gamma}_t|\mathbf{S})$ denotes the conditional estimate and $p_\phi(\mathbf{\Gamma}_t)$ is the unconditional estimate, modeled by an additional smaller denoising network $\mu_\phi$ trained specifically to model the marginal distribution of the missing values. In practice,

$$\tilde{\mu}^{num}(\mathbf{\Gamma}_t, \mathbf{S}, t) = (1 + \omega)\mu_\theta^{num}(\mathbf{\Gamma}_t, \mathbf{S}, t) - \omega\mu_\phi(\mathbf{\Gamma}_t, t). \tag{7}$$

### 3.3.2. GEOMETRIC DIFFUSION MODEL FOR BRAIN NETWORK

**Latent Diffusion Models on Manifolds of SPD and Cholesky decomposition.** Latent Diffusion Models (LDMs) (Rombach et al., 2022) operate in a compressed latent space to reduce computational complexity. However, biological data, particularly brain functional connectivity, often lies on a Riemannian manifold of SPD matrices, denoted as $\mathcal{S}_{++}^N$. Euclidean diffusion operations are ill-suited here, as they do not preserve the geometric properties of the data, i.e., adding noise to an SPD matrix does not guarantee the result is SPD. To address this, we require a framework that respects the manifold geometry, typically involving mapping data to a tangent space or utilizing decompositions to ensure valid geodesic traversing during the diffusion and sampling processes. Recent advances in geometric deep learning have sought to address this by extending diffusion and score-based generative modeling to general Riemannian manifolds (Huang et al., 2022; De Bortoli et al., 2022). In the specific context of SPD matrices, works such as SPD-DDPM (Li et al., 2024) and Riemannian Flow Matching (Collas et al., 2025) have successfully defined noise injection and denoising steps by leveraging Affine-Invariant or Log-Euclidean metrics.

Despite their theoretical rigor, these approaches typically rely on eigendecompositions or matrix logarithms that incur $O(N^3)$ complexity, posing significant challenges for high-dimensional neuroimaging data, e.g., $N = 116$ using the AAL atlas (Rolls et al., 2020). To mitigate this, alternative geometric frameworks (Lin, 2019) based on Cholesky decomposition have been explored to map SPD manifolds to computationally tractable spaces by establishing the theoretical foundation of the Log-Cholesky metric, demonstrating it as a diffeomorphism that avoids the swelling effect while remaining computationally efficient. Building on these insights, we propose leveraging a manifold-aware latent space to balance geometric fidelity with computational efficiency using Cholesky decomposition in a vector-quantized variational autoencoder (VQVAE), namely, SPD-VQVAE. The feasibility of SPD-VQVAE is guaranteed by the fact that the Cholesky decomposition acts as a one-to-one (i.e., diffeomorphism) mapping for the SPD manifold, as established in the following theorem:

**Theorem 3.1** (Uniqueness of Cholesky factor of $\mathcal{S}_{++}^N$)**.** *Let $\mathcal{S}_{++}^N$ denote the Riemannian manifold of symmetric positive definite matrices of dimension $N \times N$, and let $\mathcal{L}_{++}^N$ denote the manifold of lower triangular matrices with strictly positive diagonal entries. For $\forall M \in \mathcal{S}_{++}^N$, there is only one $L \in \mathcal{L}_{++}^N : M = LL^\top$. Consequently, the mapping $\Psi : \mathcal{S}_{++}^N \to \mathcal{L}_{++}^N$ defined by $\Psi(M) = L$ is a diffeomorphism (Golub & Van Loan, 2013).*

**SPD-VQVAE.** We first train $\mathcal{E} : \mathbf{M} \mapsto z$ and $\mathcal{D} : z \mapsto \hat{L}$, the VQVAE, which compresses the brain functional connectivity matrix into a discrete latent space while maintaining the ability to decode back to the manifolds of SPD. The encoder $\mathcal{E}$ is an MLP that flattens the input matrix $\mathbf{M}$ and projects it into a sequence of $N_q$ latent vectors $z_e \in \mathbb{R}^{N_q \times d}$, which are quantized to the nearest neighbors in a learnable codebook $\mathcal{Z} \in \mathbb{R}^{N_{code} \times d}$, yielding $z$, where $N_{code}$ is the number of codes in the codebook and $d$ is the hidden dimensionality of SPD-VQVAE. The encoder and the decoder are implemented by Multi-Layer Perceptrons (MLP).

To ensure that the reconstructed matrix $\hat{\mathbf{M}}$ is a valid SPD matrix, we do not predict $\hat{\mathbf{M}}$ directly. Instead, the decoder $\mathcal{D}$ predicts a lower-triangular matrix $L$ and reverses Cholesky decomposition $\hat{\mathbf{M}} = \hat{L}\hat{L}^\top$ after the `softplus` activation applied to the diagonal. The objective

$$\mathcal{L}_{\text{VAE}} = \underbrace{\mathcal{L}_{\text{SPD}}(L, \hat{L})}_{\text{Cholesky factor}} + \underbrace{\mathcal{L}_{\text{recon}}(\mathbf{M}, \hat{\mathbf{M}})}_{\text{Reconstruction}} \\ + \underbrace{\|\text{sg}[z_e(x)] - e\|_2^2}_{\text{Codebook Loss}} + \underbrace{\beta\|z_e(x) - \text{sg}[e]\|_2^2}_{\text{Commitment Loss}}, \tag{8}$$

where sg denotes the stop-gradient operator argmin, and $\mathcal{L}_{\text{SPD}}$ and $\mathcal{L}_{\text{recon}}$ are both using Mean Squared Error (MSE). This formulation guarantees that the generated DTs mathematically reside on the correct manifold.

**Cholesky conditional LDM.** In the second stage, we train a Denoising Diffusion Probabilistic Model (DDPM) to model the distribution of the latents $z$. The diffusion backbone is an MLP U-Net. Unlike standard U-Nets which use 2D convolutions, our model processes the sequence of latent vectors using Residual MLP Blocks (ResBlock: $z \mapsto \hat{z}$) with skip connections between the encoder and decoder paths. Conditioning is injected via two mechanisms: *(1) Time embedding.* Embed$_{\text{diff}}(t) \in \mathbb{R}^{1 \times d}$, is added to the input of each layer. *(2) Cross-attention.* The medical history context $\hat{\mathbf{y}}$ derived from the pretrained AR model $\phi$ serves as the key/value pair to update the output from ResBlock $\hat{z} = \texttt{Softmax}(QK^T/\sqrt{C_{hid}})V$:

$$Q := \hat{z}\hat{\boldsymbol{\alpha}}_h, K := \hat{\mathbf{y}}\hat{\boldsymbol{\beta}}_h, V := \hat{\mathbf{y}}\hat{\boldsymbol{\gamma}}_h, \tag{9}$$

where $\hat{\boldsymbol{\alpha}}_h \in \mathbb{R}^{d \times C_{hid}}, \hat{\boldsymbol{\gamma}}_h, \hat{\boldsymbol{\beta}}_h \in \mathbb{R}^{d_\phi \times C_{hid}}$ are learnable parameters, and $C_{hid}$ is the hidden dimensionality of the cross-attention. This allows the model to attend to specific medical history with causality information in the latent $\hat{\mathbf{y}}$ when synthesizing the DT.

Table 1. The performance comparison on the next disease prediction using different tokenization methods of EHR-to-event models, where the scores are the macro-average across all 1,944 ICD codes.

| | AUC↑ | | F1 score↑ | |
| --- | --- | --- | --- | --- |
| | GPT2 | Qwen3 | GPT2 | Qwen3 |
| Delphi | $0.6994_{\pm 0.0907}$ | $0.8931_{\pm 0.0548}$ | $7.09_{\pm 7.56}$ | $18.17_{\pm 20.67}$ |
| DiffDT | $\mathbf{0.9087}_{\pm 0.0498}$ | $\mathbf{0.9171}_{\pm 0.0486}$ | $\mathbf{18.60}_{\pm 16.29}$ | $\mathbf{20.92}_{\pm 20.40}$ |

The reported macro-average F1 scores are influenced by the long-tailed distribution of the 1,944-disease ICD vocabulary.

Table 2. The performance comparison on the next disease prediction using DT-to-event (top part) and SDoH-to-event (bottom part) methods, where scores are the average F1 for disease types of different organs.

| Diseases in | Brain | Heart | Liver | Kidney |
| --- | --- | --- | --- | --- |
| NeuroPath | $56.53_{\pm 16.78}$ | $48.96_{\pm 8.04}$ | $54.20_{\pm 19.57}$ | $54.57_{\pm 13.94}$ |
| BrainMass | $47.18_{\pm 12.02}$ | $56.63_{\pm 10.04}$ | $58.81_{\pm 16.41}$ | $50.43_{\pm 13.20}$ |
| DiffDT-Brain | $\mathbf{65.14}_{\pm 14.14}$ | $53.74_{\pm 8.35}$ | $60.44_{\pm 9.07}$ | $54.97_{\pm 18.77}$ |
| DiffDT-Heart | $58.00_{\pm 7.98}$ | $\mathbf{58.50}_{\pm 5.23}$ | $56.17_{\pm 4.02}$ | $58.18_{\pm 12.52}$ |
| DiffDT-Liver | $59.88_{\pm 7.28}$ | $52.23_{\pm 7.35}$ | $\mathbf{61.65}_{\pm 3.84}$ | $53.52_{\pm 1.81}$ |
| DiffDT-Kidney | $53.91_{\pm 2.19}$ | $54.72_{\pm 3.99}$ | $57.92_{\pm 2.82}$ | $\mathbf{64.32}_{\pm 16.92}$ |

The objective is to predict the noise $\epsilon$ added to the latent representation $z_t$

$$\mathcal{L}_{\mathrm{LDM}} = \mathbb{E}_{z \sim \mathcal{E}(\mathbf{M}), t, \epsilon, y}\left[\|\epsilon - \epsilon_\theta(z_t, t, \hat{\mathbf{y}})\|^2\right] \quad (10)$$

where $\epsilon_\theta$ is the MLP U-Net. During inference, we sample a latent $z_0$ conditioned on the patient's history and decode it using the frozen SPD-VQVAE decoder to obtain the generative biomarkers.

## 4. Experimental Results

Our experiments are designed for a thorough evaluation: **(1) The next disease prediction** for different past-future disease pairs and **(2) different organs** as the mediation inference. **(3) The topological DT generation** by Cholesky LDM using the proposed SPD-VQVAE.

**Dataset Setup.** The UKB dataset (Bycroft et al., 2018) is used in this work. Our experiments utilized 44,834 brain functional connectivity matrices, 23,987 heart, 32,155 kidney, and 28,722 liver tabular samples from 20,057, 23,987, 32,155, and 28,722 subjects, respectively. The tabular traits are derived from the corresponding imaging data (see Appendix A for details). Every tabular and topological biomarker is associated with an ICD sequence, the sequence of ages being diagnosed, and the imaging date. In total, the train:val ratio is set as 80%:20% at subject level, and this subject-level split is strictly maintained across all training phases (AR, diffusion, and predictive finetuning) so that no subject in the validation set is ever observed during training of any model component, eliminating data leakage. The universal subject IDs of the validation set are released alongside our code repository to support reproducibility. AR

Table 3. The performance comparison on tabular DT generation.

| | Heart (112 traits) | Liver (3 traits) | Kidney (3 traits) |
| --- | --- | --- | --- |
| RMSE↓ | $0.265_{\pm 0.003}$ | $0.184_{\pm 0.018}$ | $0.146_{\pm 0.015}$ |
| WD↓ | $17.265_{\pm 33.814}$ | $2.477_{\pm 3.202}$ | $0.991_{\pm 0.313}$ |

Table 4. The comparison on topological DT generation in the context of medical history, where SPD-V. denotes for SPD-VQVAE and S.-V.-Dual is SPD-VQVAE-Dual, and mAcc is the mean accuracy for topology prediction, i.e., the binary classification if there is an edge between two brain regions using various thresholds $\in$ [0.5, 0.9] with a stride of 0.1.

| LDM | RMSE↓ | WD↓ | $r\uparrow$ | mAcc↑ |
| --- | --- | --- | --- | --- |
| w/ VQVAE | $0.261_{\pm 0.027}$ | $7.110_{\pm 0.461}$ | $0.503_{\pm 0.054}$ | $90.87_{\pm 0.10}$ |
| w/ SPD-V. | $0.220_{\pm 0.034}$ | $6.019_{\pm 0.712}$ | $0.677_{\pm 0.077}$ | $95.71_{\pm 0.18}$ |
| w/ S.-V.-Dual | $\mathbf{0.203}_{\pm 0.031}$ | $\mathbf{5.841}_{\pm 0.704}$ | $\mathbf{0.726}_{\pm 0.072}$ | $\mathbf{98.36}_{\pm 0.10}$ |

models, Delphi and DiffDT, are trained with 7,276,575 ICD tokens from 448,651 subjects. We use the level-2 ICD code, resulting in a vocabulary of 1,944 diseases, where 964 of them appeared for subjects with sensory data. While the majority of UKB imaging is cross-sectional, the validation set ($n = 3,425$) includes a longitudinal subset with two imaging timepoints (Brain: $n = 188$, Heart: $n = 212$, Kidney: $n = 98$, Liver: $n = 229$) that supports temporal evaluation of generated DTs.

**Baseline Setup.** Delphi (Shmatko et al., 2025) is an AR model present recently, explicitly learning the causality between ICD events to predict multi-morbidity along the longitudinal EHR data in the UKB dataset. Therefore, Delphi is the main competitor for next disease prediction to compare their age embedding-derived tokenization with our adaptive tokenization. Furthermore, a generalizability test of our method on a modern state-of-the-art (SOTA) AR model is appropriate since Delphi originally uses GPT2 (Radford et al., 2019). Qwen3 (Yang et al., 2025) is an open-source SOTA AR architecture in the field of language generation and is utilized to replace GPT2. In addition to the EHR-to-event model, brain network models that remap the imaging biomarkers to latent space for a better representation are considered as DT-to-event models (Yang et al., 2024; Wei et al., 2024; 2025a;b). While most of them tested on limited diseases, e.g., Alzheimer's and Parkinson's diseases, the pretrained BrainMass (Yang et al., 2024), and NeuroPath (Wei et al., 2024) are selected for comparison.

**Implementations.** The backbone architecture of Choleksy conditional LDM can be found in Appendix B. Since there is evidence (Chen et al., 2022) showing that the topological biomarker, i.e., brain network, shares latent features, which leads LDM to generate similar brain functional connectivity matrices with no personalized pattern, we implement the proposed Cholesky LDM with two branches of SPD-VQVAE (SPD-VQVAE-Dual). Two branches generate low-passed and high-passed brain functional connectivity matrices using the Fourier transformation with a threshold

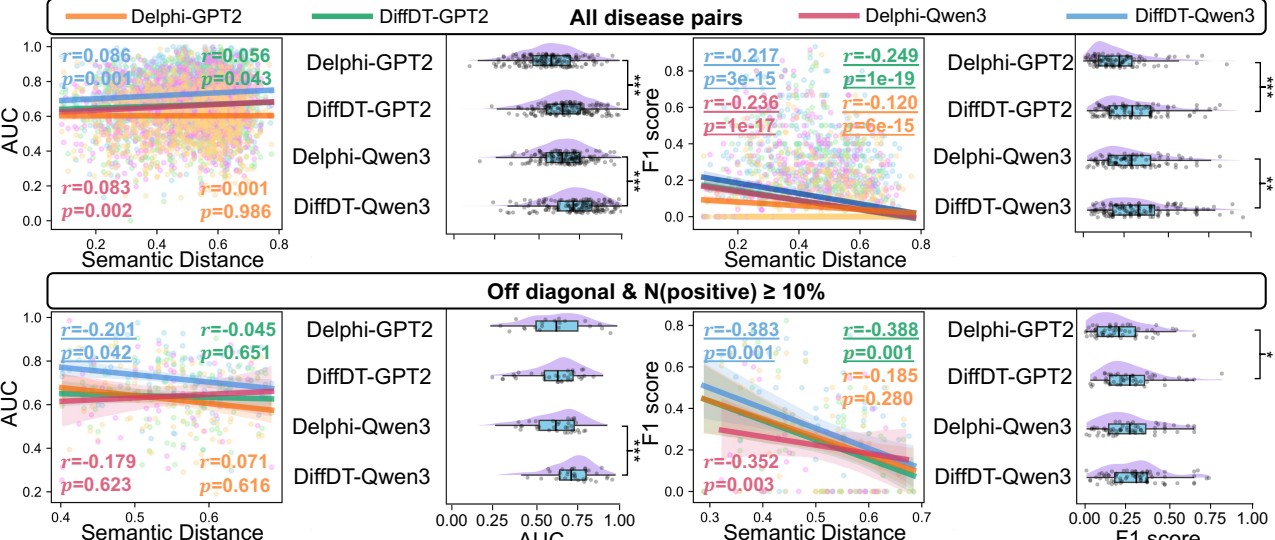

*Figure 4.* The performance comparison on the next token prediction with respect to the semantic distance between the past and the future disease at the imaging date. AUC and F1 scores are calculated for each disease, the correlation is denoted by Pearson's $r$ and $p$ value, and the significant improvement are marked by $*: 1e-3 < p < 0.05$, $**: 1e-10 < p < 1e-3$, and $***: p < 1e-10$ using the $t$-test. Note that $r$ and $p$ with a significant negative correlation are underlined.

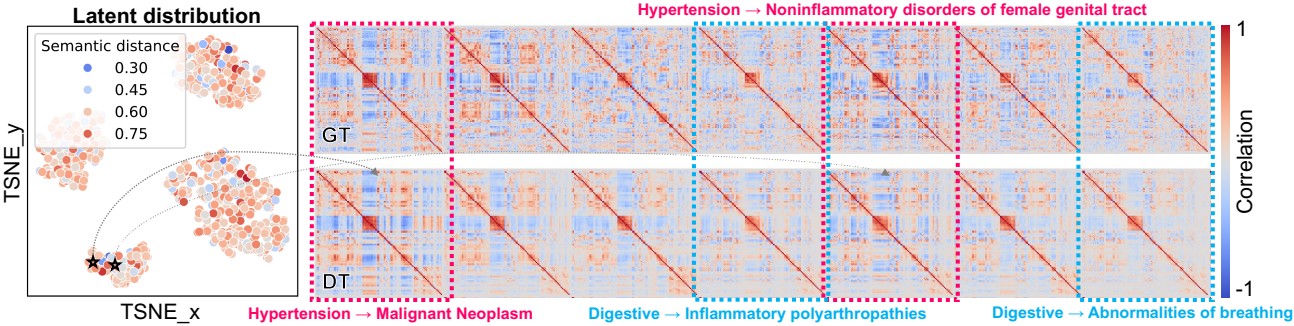

*Figure 5.* The qualitative results of topological DT generation by *DiffDT* in the context of medical history. **Left** is the TSNE distribution (Maaten & Hinton, 2008) of generated latent features by conditional Cholesky LDM. **Right** is the ground truth (GT) and the digital twin (DT) decoded from the generated latent feature, where brain functional connectivity matrices with the same past disease are marked with dashed boxes.

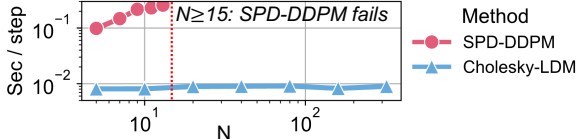

*Figure 6.* Runtime comparison on one diffusion step between existing SPD-DDPM (Li et al., 2024) and our Cholesky LDM.

set to 25, separately. See Appendix C for ablation studies. The predictive model of *DiffDT* for topological DT is pretrained BrainMass. For tabular DT, we use a standard Transformer encoder. See Appendix E for details of predictive models. Following (Shmatko et al., 2025), multiclass F1 and AUC scores, i.e., taking one disease as positive and others as negative, are used as metrics.

**Accuracy of next disease prediction.** As listed in Table 1, we compare *DiffDT* with EHR-to-event models using different AR architectures. The average scores for all diseases show a performance boost when using *DiffDT*, whether

based on GPT2 or Qwen3. Although F1 scores have only 18% to 20% for *DiffDT*, the next disease is sampled based on the predicted logits, which holds a great performance by the AUC above 0.9.

As shown in Fig 4, we calculate AUC score within the same past-future disease pair since the performance degrades when the semantic distance between a pair increases in Fig 2. AUC and F1 scores versus the distance demonstrate a similar upgrade after using *DiffDT*, where the performance improvement is significant for all disease pairs, as well as Qwen3 and GPT2 for AUC and F1 of the off-diagonal pairs, respectively. We can observe that the performance pitfall of Delphi is mainly caused by zeros in F1 scores.

As listed in Table 2, we compare *DiffDT* using different organs with DT-to-event models to show the proposed SDoH-to-event performance. Overall, *DiffDT* always holds the best performance for the same organ used as the mediation infer-

ence. Although *DiffDT*-Brain and BrainMass used the same pretrained predictive model, the generated DT conditioned on ICD-coded SDoH proxies as the input outperformed the GT biomarker by nearly 18% F1 score, indicating the multi-pathway probabilistic mediation learned by *DiffDT* is beneficial for the next disease prediction. We note that Table 2 reports F1 on organ-specific disease categories where models achieve performance in the 50–65% range, which is state-of-the-art for multi-label next-token prediction over a 1,944-disease vocabulary (Shmatko et al., 2025).

**DT Generation Performance.** As listed in Table 3, the tabular DT generation is evaluated by Root Mean Square Error (RMSE) and Wasserstein distance (WD) per normalized trait for three organs. Given lower errors higher the F1 in Table 2, tabular DT generative performance is correlated to the performance as the organ mediation in *DiffDT*.

To demonstrate our contribution of SPD-VQVAE, we list the topological DT generation comparison in Table 4. It is clear that SPD-VQVAE-Dual performed the best. The qualitative results in Fig. 5 show that personalized patterns are well maintained, while VQVAE or SPD-VQVAE tends to diminish details in Appendix Fig. 8.

**Computational Complexity.** As shown in Fig. 6, the efficient computational complexity of our Cholesky LDM makes *DiffDT* feasible. End-to-end, generating one digital twin and predicting the next disease takes approximately 1.2 seconds per subject per mediation on a single NVIDIA RTX 6000 Ada GPU (48 GB), scaling linearly to 5.6 seconds for five sequential mediations. This is supported by our Cholesky LDM, which avoids the $O(N^3)$ complexity of standard SPD-manifold operations.

**Backbone scaling vs. *DiffDT* contribution.** To verify that the gains from *DiffDT* are not advanced by stronger AR backbones, we additionally trained Delphi and *DiffDT* with a Qwen3.5 backbone. Even with this stronger backbone, integrating *DiffDT* continues to provide a +0.12 AUC improvement over Delphi-Qwen3.5, indicating that the contribution of digital-twin mediation is complementary to and largely independent of LLM scaling.

**Counterfactual evaluation.** To test whether *DiffDT* generates biologically meaningful counterfactual digital twins, we follow (Rasal et al., 2024) and replace an exposure ICD code (e.g., $X$=G12) in a patient's history with a `[Healthy]` token, then generate the resulting do(healthy) DT. We compare it to ground-truth (GT) DTs from real diseased and real healthy subjects using Fréchet Inception Distance (FID), Wasserstein Distance (WD), and Pearson correlation $r$. As reported in Table 5, do(healthy) DTs are significantly closer to GT healthy than to GT diseased on both FID ($p$=2.5e−5) and WD ($p$=1.6e−6), confirming that the intervened DTs reflect the targeted counterfactual physiological state.

*Table 5.* Counterfactual DT evaluation.

| | FID↓ | WD↓ | $r$ ↑ |
|---|---|---|---|
| do(healthy) vs. GT$_{\text{disease}}$ | $92.96_{\pm 8.67}$ | $8.95_{\pm 0.24}$ | $0.39_{\pm 0.07}$ |
| do(healthy) vs. GT$_{\text{healthy}}$ | $\mathbf{51.37}_{\pm 4.13}$ | $\mathbf{6.91}_{\pm 0.23}$ | $0.40_{\pm 0.08}$ |
| $p$-value | 2.5e-5 | 1.6e-6 | 0.148 |

*Table 6.* Mean absolute ATE error (↓) across 10 exposure-outcome ICD pairs. Per-pair results in Appendix Table 7.

| | Delphi | *DiffDT* |
|---|---|---|
| Mean ATE Abs. Error↓ | $0.007_{\pm 0.005}$ | $\mathbf{0.004}_{\pm 0.004}$ |

**Probabilistic mediation evidence.** We further estimate the Average Treatment Effect (ATE) following (Louizos et al., 2017): a propensity-score MLP is fit on the generated DT for *DiffDT* (and on Delphi token embeddings for the baseline), and the empirical ATE is computed from the multi-label binarized ICD sequence. As summarized in Table 6, *DiffDT* more than halves Delphi's mean absolute ATE error across the ten exposure-outcome ICD pairs, supporting the interpretation of generated DTs as biologically meaningful mediators while remaining within the scope of associative modeling on observational data; per-pair results are in Appendix Table 7.

## 5. Conclusions and Limitations

In this work, we proposed a disease reasoning framework, *DiffDT*, based on a new generative objective, the digital twin (DT) conditioned on ICD-coded proxies of social determinants of health as the probabilistic mediation inference connecting past and future diseases. *DiffDT* brought the central role of sensor-derived diagnosis to autoregressive next disease prediction using the diffusion of multi-organ DT in the context of healthcare history, filling the gap of personalized disease reasoning. By proposing an efficient Cholesky LDM for topological DT and integrating with a new ICD tokenization method, *DiffDT* showed the best performance on disease prediction and DT generation.

**Limitations.** (1) UK Biobank exhibits healthy volunteer bias, so *DiffDT* predictions should be recalibrated before deployment in epidemiologically distinct cohorts. (2) UKB imaging is predominantly cross-sectional, so individual-level disease trajectories rely on population-level interpolation rather than per-subject longitudinal evidence. (3) The gain from the DiffDT mediation module is modest relative to scaling the AR backbone, though *DiffDT* still adds a consistent boost on Qwen3 and Qwen3.5. (4) "SDoH" here refers only to ICD-coded Z and V–Y proxies; the model ingests no direct social, economic, behavioral, or environmental measurements. A genuine social-determinants digital twin would require integrating high-resolution multimodal data, which is our most important next step.

## Acknowledgements

This work was supported by the National Institutes of Health (AG091653, AG068399, AG084375) and the Foundation of Hope.

## Impact Statement

This paper presents work whose goal is to advance the field of Machine Learning. There are many potential societal consequences of our work, none which we feel must be specifically highlighted here.

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

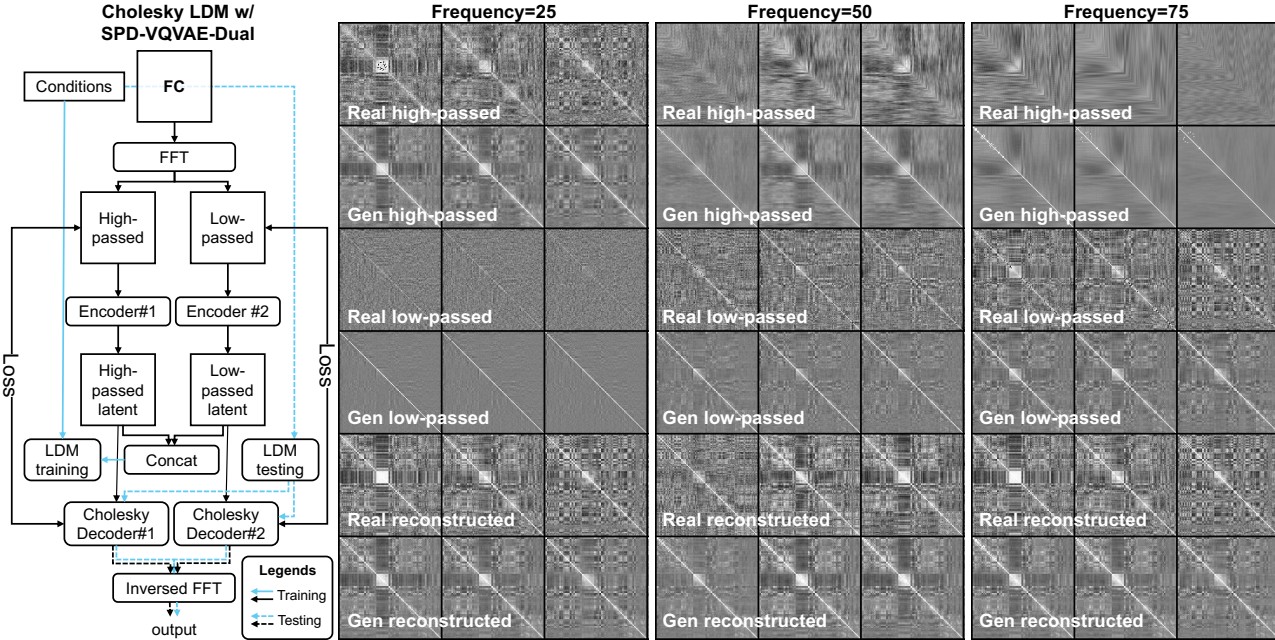

*Figure 7.* **Left**: The pipeline of training and testing the proposed Cholesky LDM with SPD-VQVAE-Dual. **Right**: The qualitative results of real and generated ('Gen') brain functional connectivity matrices using different frequency thresholds for Fourier transformation to decompose the matrix.

## A. Data and Codes

The code of *DiffDT* can be found in this URL: `https://github.com/Chrisa142857/DiffDT`. The brain functional connectivity matrices are extracted from brain functional MRI (fMRI) using fMRIPrep (Esteban et al., 2019), a widely adopted and fully standardized pipeline, URL: `https://fmriprep.org/en/stable/`. The structural connectivity (SC) matrices are extracted from brain Diffusion Weighted Imaging (DWI) as the input of NeuroPath (Wei et al., 2024) using QSIprep (Cieslak et al., 2021), an established diffusion pipelines, URL: `https://qsiprep.readthedocs.io/en/latest/`.

The tabular multi-organ imaging traits of heart, liver, and kidney are extracted from MRI data and released in the UKB dataset. The traits with field IDs under category ID 162, 159, and 126, except for the protocol-related fields, are manually selected in the URL `https://biobank.ndph.ox.ac.uk/ukb/`, where each field ID represents a phenotypic feature from image preprocessing results, e.g., the ascending aorta distensibility. Only subjects with complete traits for each organ are used in the experiments.

The level two ICD description is extracted from URL `https://biobank.ndph.ox.ac.uk/ukb/field.cgi?id=41270`. Then, those text descriptions are converted to text embeddings using the latest Qwen3 `https://ollama.com/library/qwen3-embedding:latest` for semantic distance analysis.

The ICD codes of diseases for different organs are selected following (Hussein et al., 2022) (brain), `https://www.cms.gov/medicare/coding/icd10/downloads/icd10clinicalconceptscardiology1.pdf` (heart), the top level description (N00-99 for kidney), and the level two description (K70-77 for liver).

## B. Cholesky LDM architecture

The MLP U-net of Cholesky LDM has Up, Middle, and Down parts. Each part has three ResBlocks, where the hidden dimensionalities of Up, Middle, and Down are increasing 256→512→1024, fixed, and decreasing 1024→512→256, respectively. The output from every ResBlock in Down is connected to the corresponding one in Up, the same as CNN U-net (Ronneberger et al., 2015). The pipeline of training and testing Cholesky LDM with SPD-VQVAE-Dual is shown in Fig. 7 left, where encoder#1 and encoder#2 are both a three-layer MLP with decreasing hidden dimensionalities (512→256→128),

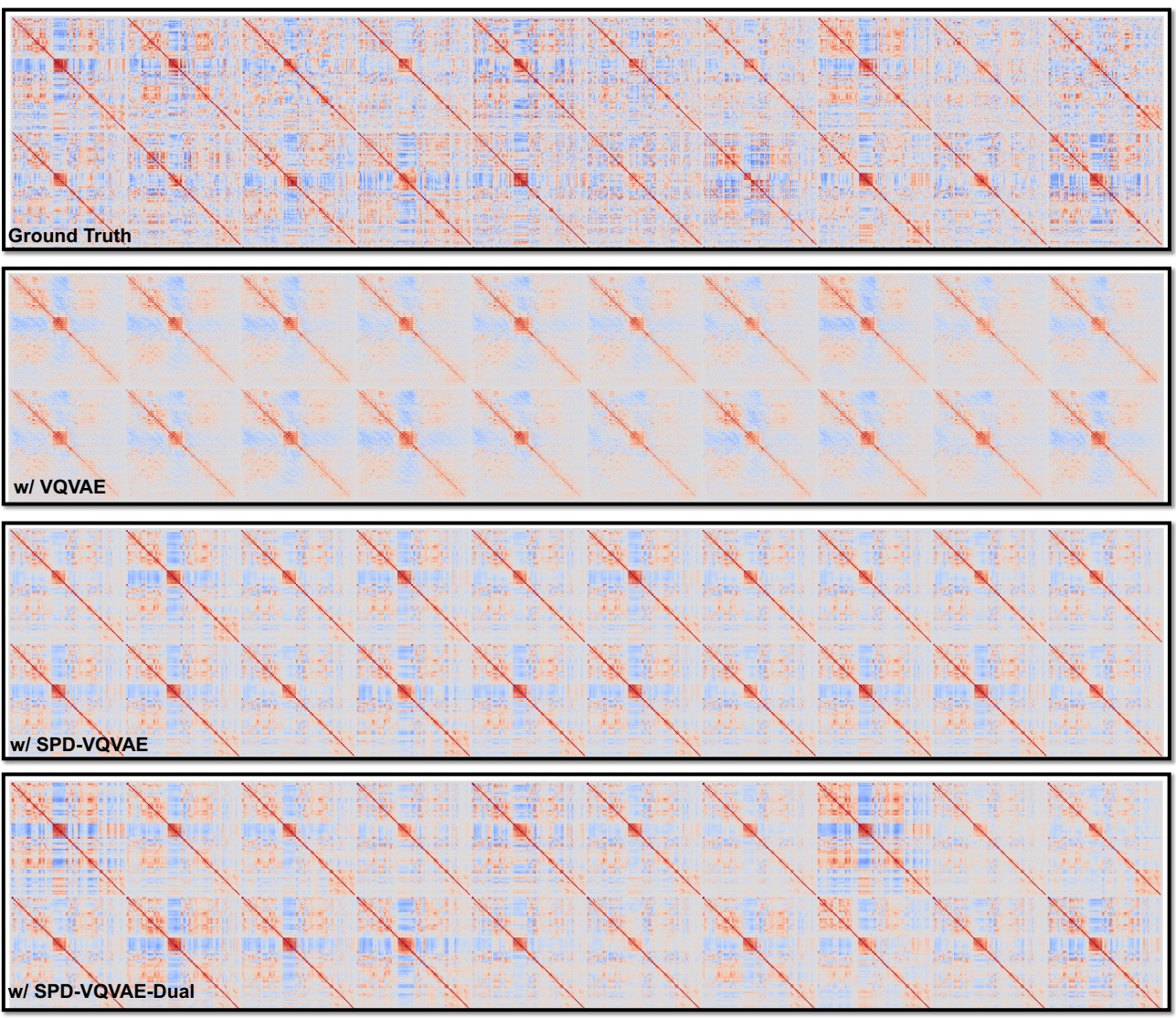

*Figure 8.* Qualitative comparison on conditional Cholesky LDM using different autoencoding methods.

resulting in an information bottleneck. Networks in Cholesky decoder#1 and decoder#2 are also a three-layer MLP with reverse hidden dimensionalities (128→256→512).

## C. Abaltion studies of SPD-VQVAE-Dual

The core idea of SPD-VQVAE-Dual is to decompose the personalized and shared features in the brain functional connectivity matrix according to their frequencies. As shown in Fig. 7 right, we tested three difference frequency threshold, 25, 50, and 75, and observe the decomposed matrices to ensure the pattern of one is the most similar to others and another one has the most diverse pattern. Although the high-passed matrices have a diverse global pattern, i.e., the cross shape, when frequencies are set to 50 and 75, the blocking patterns are also filtered out to the low-passed matrices. This causes that blocks in off-diagonal area can be only observed in the generated reconstructed matrices when using frequency=25. Therefore, we choose 25 as the frequency threshold for SPD-VQVAE-Dual

*Table 7.* Per-pair Average Treatment Effect (ATE) estimation results. Summary numbers (last row) appear in the main text Probabilistic mediation evidence paragraph.

| $T{\to}Y$ | $N_{exp}$ | $N_{out}$ | ATE | | | ATE Abs. Error↓ | |
|---|---|---|---|---|---|---|---|
| | | | Empirical | *DiffDT* | Delphi | *DiffDT* | Delphi |
| I10→I25 | 902 | 238 | 0.089 | 0.089 | 0.073 | **0.000** | 0.016 |
| I10→K57 | 902 | 179 | 0.041 | 0.043 | 0.044 | **0.002** | 0.003 |
| I10→E78 | 902 | 142 | 0.101 | 0.114 | 0.097 | 0.013 | **0.004** |
| I10→K21 | 902 | 147 | 0.069 | 0.070 | 0.071 | **0.001** | 0.002 |
| I10→R07 | 902 | 133 | 0.043 | 0.050 | 0.050 | 0.007 | 0.007 |
| I10→I48 | 902 | 130 | 0.050 | 0.049 | 0.042 | **0.001** | 0.008 |
| I10→K29 | 902 | 129 | 0.033 | 0.036 | 0.037 | **0.003** | 0.004 |
| K57→K64 | 502 | 124 | 0.036 | 0.043 | 0.025 | **0.007** | 0.011 |
| I10→M19 | 502 | 124 | 0.039 | 0.043 | 0.052 | **0.004** | 0.013 |
| I10→I20 | 902 | 114 | 0.036 | 0.040 | 0.035 | 0.004 | **0.001** |
| Avg | – | – | – | – | – | **$0.004_{\pm 0.004}$** | $0.007_{\pm 0.005}$ |

## D. Qualitative comparison on conditional Cholesky LDM

As shown in Fig. 8, the qualitative comparison on conditional Cholesky LDM demonstrates the contribution of the proposed SPD-VQVAE. VQVAE as the conventional method cannot produce symetric matrices, while it can keep the global shape of a brain functional connectivity matrix with a blue cross and a red block, where those are nodes in the Visual Network (VN) showing negative correlation with other networks but highly correlated to each other within VN. SPD-VQVAE makes better generations with SPD matrices, but the difference between most of brain functional connectivity matrices is hard to find. In contrast, SPD-VQVAE-Dual produces the best matrices while maintaining them on the manifold of SPD.

## E. Predictive models

For the topological DT, we use BrainMass (Yang et al., 2024) with an MLP predictive head for the next disease prediction. BrainMass is the brain foundation model with the lightest weight (34M) comparing to others, leading to no delay for the mediation inference. BrainMass is pretrained with unsupervised learning on UKB (Bycroft et al., 2018), HCP-aging (Bookheimer et al., 2019), and HCP-young adult (Van Essen et al., 2013) datasets, containing 68,251 brain functional connectivity matrices. Then, it is finetuned on the real brain functional connectivity matrices or brain functional connectivity matrices generated by *DiffDT*-Brain to demonstrate DT-to-event or SDoH-to-event performance, respectively, in the Table 2.

For the tabular DT, we build a six-layer eight-head Transformer encoder with $\text{Input} = [\text{CLS}, \text{Trait}]$ as the input, where 'CLS' denotes a randomly initialized classification embedding that handles the next disease classification, and 'Trait' denotes columns in the generated tabular DT. Note that 'Trait' is the discrete version of the tabular DT by rounding the continuous value.

## F. Per-pair Average Treatment Effect estimation

Table 7 reports the full per-pair ATE estimates summarized in the main text. For each exposure-outcome ICD pair $T{\to}Y$, we report the number of exposed subjects ($N_{\text{exposure}}$), the number of subjects with the outcome ($N_{\text{outcome}}$), the empirical ATE computed directly from the multi-label binarized ICD sequence, the ATE estimated by *DiffDT* (via a propensity-score MLP fit on the generated DT) and by Delphi (via token embeddings), and the absolute error of each model against the empirical ATE. The candidate pairs are restricted to ICD pairs with sufficient subjects experiencing both exposure and outcome, given the long-tailed disease distribution.

