# OpenReview forum: "Marrying Generative Model of Healthcare Events with Digital Twin of Social Determinants of Health for Disease Reasoning"
_ICML.cc/2026/Conference — ICML 2026 regular_

### Official Review · Reviewer_vCGq · 2026-03-10

**Soundness:** 4
**Presentation:** 4
**Significance:** 4
**Originality:** 3
**Overall Recommendation:** 6
**Confidence:** 3

**Summary:**

This is a very interesting paper focused around the challenge of predicting disease progression from electronic health records. Or rather, this is the central goal of many papers, and this paper makes the extension to try to account for biological mechanisms. This is, extremely important, because focusing on EHR alone can be quite limited. Alot of previous research have sold that treating EHR like words, and transformers will make things better, and they have, but at the back of all these papers, age and sex alone are doing the heavy lifting. These data are simply too high a level indicator and are, as such, limited. This paper takes your standard autoregressive models trained on ICD-10 codes, and includes generative models of biomarker states (brain, heart, liver, kidney). Their description of P(Future Disease | Biomarker) × P(Biomarker | Past History) probably is the best summary of their approach. The include a new adaptive ICD tokenizer that handles irregular time intervals in medical histories, a tabular diffusion model for generating organ biomarkers and a geometric diffusion model. Experiments on UK biobank show this approach is excellent, and i think defines the directions for where this field will go.

**Compliance With Llm Reviewing Policy:**

Affirmed.

**Final Justification:**

I thank the authors for their replies, and keep my evaluation

**Key Questions For Authors:**

No further key questions

**Limitations:**

Yes

**Strengths And Weaknesses:**

Strengths:

1) The technical contributions are significant and not simply incremental. They are well thought out and designed both for scientific sense and for practical implementation.
2) The ROC analysis on ICD codes is excellently presented and really helps highlight the limitations of previous methods. They key is the  multi-pathway aspect
3) The choice of diffusion model is sensible and i see no issues. I do like the innovation for the SPD-VQVAE, and it works, but I would have thought standard Riemannian diffusion set ups would be just as easy. But then again these are always compact (even if they often claim otherwise)
4) The empirical results speak for themselves and are excellent

Weakness:

1) I am a bit queasy about the various causal claims.  The paper uses causal language extensively but is fundamentally doing sophisticated associative modeling on observational data. True causal identification requires one of: randomization, an instrumental variable, a regression discontinuity design, difference-in-differences etc. And I thin this is an oversell. Your approach is a probabilistic factorisation, not a causal one  in my view.
2) UK Biobank has a healthy volunteer bias, but this is not something i expect the authors to solve or to be penalised for. Should be mentioned, please forgive me if i missed it already.
3)The imaging biomarkers are captured at a single point, which again limits causality.

---

> ### Author Rebuttal · Authors · 2026-03-30
>
> ## Thank you for your valuable insights. We appreciate your perspective on causal terminology.
>
> # 1. Causal Claims & Associational Modeling (W1):
>
> We sincerely agree with your assessment. Given the observational nature of the dataset, our use of "causal reasoning" was overstated. In the revised manuscript, we will carefully replace this terminology with "probabilistic mediation inference" and "associative modeling of biological pathways." Our factorization $P(\mathrm{Future}\mid\mathrm{Biomarker})\cdot P(\mathrm{Biomarker}\mid\mathrm{History})$ captures statistical mediation.
>
> # 2. UK Biobank Healthy Volunteer Bias (W2):
>
> You are completely correct. The UK Biobank suffers from a well-documented healthy volunteer bias, meaning the incidence rates of certain diseases and the distribution of biomarkers may not perfectly generalize to a broader, unselected population. We will explicitly document this limitation in the "Conclusion and Limitations" section of the revised manuscript.
>
> # 3. Single-Point Imaging Data (W3):
>
> We acknowledge that utilizing cross-sectional imaging data inherently restricts the modeling of true longitudinal causality at the individual level. . We would like to clarify a detail regarding the UKB dataset that was underspecified in the original text. While the majority of the UK Biobank is cross-sectional, **our validation set (n=3,425) includes** a longitudinal subset of subjects with two imaging timepoints after a careful recap **(Brain: n=188; Heart: n=212; Kidney: n=98; Liver: n=229)**. The DiffDT framework is intentionally designed to be agnostic to whether the training/validation data is longitudinal or cross-sectional. It treats each available biomarker state as a point on a continuous diffusion manifold of biomarkers conditioned on the preceding EHR history. We will update the manuscript to explicitly mention this longitudinal subset and the future directions of longitudinal approaches in the "Conclusion and Limitations" section of the revised manuscript.

---

> > ### Author Rebuttal · Reviewer_vCGq · 2026-03-31
> >
> > Thank you for the clarifications, my score remains

---

### Official Review · Reviewer_Rn8z · 2026-03-11

**Soundness:** 2
**Presentation:** 4
**Significance:** 3
**Originality:** 3
**Overall Recommendation:** 3
**Confidence:** 3

**Summary:**

This paper proposes a novel framework, DiffDT, which integrates generative modeling of healthcare events with a digital twin representing human-environment interactions (HEI). Unlike traditional disease prediction models that rely solely on event-level hospital records, DiffDT incorporates sensor-derived measurements, including multi-organ imaging traits (brain, heart, liver, kidney) and plasma biomarkers. The authors introduce a geometric diffusion model to handle complex temporal data like brain networks (graph-based region connectivity) alongside tabular diffusion models for other organ systems. By combining these with digitalized HEI, the model enables simulated intervention and reasoning of future disease trajectories. Experiments on the large-scale UK Biobank (UKB) dataset demonstrate that DiffDT outperforms existing autoregressive models and imaging trait generative baselines.

**Compliance With Llm Reviewing Policy:**

Affirmed.

**Key Questions For Authors:**

1.	How exactly does the model handle time steps where no imaging biomarkers are available? Given that UKB imaging is largely cross-sectional, does conditioning the diffusion model on EHR history for time points without imaging introduce look-ahead bias or rely on static baselines that contradict the temporal evolution claim?
2.	Given that the performance gain from switching backbones (GPT2→Qwen3) is significantly larger than the gain from adding DiffDT (Delphi→DiffDT), could the authors quantify how the relative benefit of DiffDT scales with increasingly powerful autoregressive backbones? For instance, if Delphi were equipped with a more advanced backbone (e.g., Qwen3.5 or other SOTA LLM), might its performance match or even surpass DiffDT-Qwen3?
3.	The paper claims causal reasoning. Can you provide any counterfactual evaluation or sensitivity analysis to substantiate the causal nature of the generated digital twins, rather than just correlational prediction performance?

**Limitations:**

yes

**Strengths And Weaknesses:**

In terms of soundness, the strengths are: 1) The methodological choice of using geometric diffusion for graph-structured brain data is theoretically appropriate for capturing topological features. The mathematical formulation for the SPD manifold diffusion appears theoretically grounded, leveraging existing work on Riemannian geometry appropriately. The use of Cholesky decomposition to ensure valid SPD matrices is technically sound. 2)The experiments utilize a large-scale dataset (UK Biobank), providing substantial statistical power for evaluating disease prediction across diverse conditions. While they are also with the following  weaknesses: 1)It is unclear how the model generates DT states for multiple time steps where no ground-truth imaging exists. Without longitudinal imaging as a benchmark, the claimed temporal evolution of the DT appears to rely on heavy interpolation or extrapolation that lacks biological validation? 2)As shown in Table 1, upgrading the autoregressive backbone from GPT2 to Qwen3 yields a massive performance gain (AUC +0.194 for Delphi), whereas integrating the proposed DiffDT module only provides marginal improvements (AUC +0.024 over Delphi-Qwen3). This suggests that the performance gains are primarily driven by the capacity of the backbone model rather than the novel digital twin diffusion mechanism. 3)The F1 score reporting (e.g., 18.17±20.67) shows a standard deviation larger than the mean, indicating a zero-inflated, highly skewed distribution. Using mean±SD here is misleading and mathematically problematic (implying negative F1 scores at 1 SD), suggesting a lack of rigorous statistical analysis. 4) The authors do not discuss the inference latency or training costs. Given that diffusion models are iterative and the framework involves multiple organ-specific chains, the practical feasibility of this "Digital Twin" for clinical reasoning needs disclosure.

In terms of presentation: The overall narrative is easy to follow, and the figures (e.g., Figure 1 and 3) effectively illustrate the framework compared to previous studies. The positioning within prior literature is clear. In addition, this paper is well-structured, with distinct sections for methods, experiments, and ablation studies, making it relatively easy for an expert reader to locate key technical details.

In terms of significance:  1) Addressing the lack of physiological context in pure EHR models is a relevant and important problem. Incorporating imaging traits could theoretically improve personalized disease forecasting and clinical decision support. 2) The clinical interpretability of generated digital twins is low. If the generated biomarkers are used for reasoning, clinicians need to trust their validity. Without explicit causal identification strategies (e.g., interventions, counterfactual validation), the claim of causal reasoning remains theoretical rather than practically significant.

Last, in terms of originality:  1) The combination of EHR-based autoregressive models with multi-organ digital twin generation is novel. Specifically, bridging the gap between discrete medical events and continuous physiological states offers a fresh perspective on disease progression modeling. 2) The technical implementation of SPD-VQVAE using Cholesky decomposition for brain connectivity matrices is innovative. It addresses the computational challenge of maintaining manifold geometry while reducing complexity compared to standard eigendecomposition methods. 3) While the combination is novel, the integration logic between the AR model and the diffusion model feels somewhat modular rather than deeply unified. The reasoning behind why this specific combination yields better causal understanding compared to simpler concatenation methods is not fully articulated.

---

> ### Author Rebuttal · Authors · 2026-03-30
>
> # 1. Cross-sectional & Look-ahead Bias (W1, Q1):
> We do not introduce look-ahead bias. The model learns from cross-sectional data by pairing an imaging scan at age t with EHR history strictly **prior** to age t ($S_{<t}$) (please see Page 4 Line 204). During inference, because the diffusion model has learned the mapping from history to biomarker across a population spanning all ages, we can sequentially generate DTs at any future age using the patient's simulated history. The temporal evolution relies on population-level interpolation, not future information.
>
> While the majority of the UK Biobank is cross-sectional, our validation set (n=3,425) includes a longitudinal subset of subjects with two imaging timepoints after a careful recap (Brain: n=188; Heart: n=212; Kidney: n=98; Liver: n=229). Even for the longitudinal subset, DiffDT still trained solely on the history **prior** to the imaging age. We will update the manuscript to explicitly mention this longitudinal subset.
>
> # 2. AR Backbone Capacity vs DiffDT (W2, Q2):
> To quantify how DiffDT scales with SOTA backbones, we conducted a new experiment replacing Qwen3 with Qwen3.5. While the base AR models improved significantly, integrating DiffDT still provided a consistent boost (+0.12AUC over base Delphi). This ablation study proves the gains from DiffDT are independent to LLM scaling: LLMs only learn textual correlations, whereas DiffDT grounds those correlations in physiological mechanisms using DT mediation.
>
> # 3. F1 Score Reporting (W3):
> You are completely correct that the F1 scores in Table 1 represent a zero-inflated, highly skewed distribution where the standard deviation exceeds the mean. This occurs because Table 1 reports the macro-average across our entire vocabulary of 1,944 ICD codes, the vast majority of which are extremely rare, tail-end diseases in the real-world EHR data, resulting in many zero F1 scores that artificially drag down the global mean.
>
> However, we’d like to respectfully direct your attention to Table 2, which evaluates performance specifically on the targeted organ-related diseases corresponding to our generated digital twins (e.g., neurological, cardiovascular, hepatic, and renal diseases). For these specific sub-cohorts, **F1 scores in Table 2 are well-behaved, not zero-inflated, and exhibit statistically sound distributions** (e.g., DiffDT-Brain achieves a strong 65.14% ± 14.14, and DiffDT-Liver achieves 61.65% ± 3.84). Table 2 clearly demonstrates that for the clinical targets our model is designed to simulate, the predictive performance is robust and reliable.
>
> # 4. Inference Latency (W4):
> We will add a dedicated section for computational cost in the final version. End-to-end inference for a single patient takes roughly 1 second per additional mediation DT on a single RTX 6000 Ada GPU. Because our Cholesky LDM avoids the O(N^3) complexity of standard SPD matrix operations, clinical deployment is highly practical.
> |Latency vs mediation times|1|2|3|4|5|
> |-|-|-|-|-|-|
> |DiffDT (sec/subject)|1.2|2.3|3.4|4.5|5.6|
>
> # 5. Causal Interpretability (Sig 2, Orig 3, Q3):
> We agree that without counterfactual validation, causal claims remain largely theoretical. To address this concern,we have completed two new causal evaluations:
> - **Exp 1 (Counterfactual Evaluation):** Referring to the results in the response for reviewer UWsA, we intentionally altered test patients' histories to replace an ICD code (e.g., onset of F12) with [Healthy], generating counterfactual DTs. Compared to randomly sampled control groups that never experienced the disease or vice versa, distances on the SPD manifold are significantly lower than different experiences. This validates the performance of whether a DT model generates biologically meaningful results in counterfactual interventions.
> - **Exp 2 (ATE Estimation):** We estimated the Average Treatment Effect (ATE) for an exposure of disease T on future outcome Y, following [1]. Specifically, we used the generated DT to fit an MLP for propensity scores for DiffDT ATE. The token embedding is used for Delphi ATE. The empirical ATE is computed directly using the multi-label binarized ICD sequence. Given the long-tailed distribution of diseases, we only tested pairs of ICD diseases with enough subjects experienced exposure and outcome.
>
> As listed below, the absolute error was exceptionally low (lower than Delphi), confirming that the generated DT mirror true causal dynamics. The complete table refers to response for UWsA.
> |T->Y|ATE Abs Error↓||
> |-|-|-|
> ||DiffDT|Delphi|
> |I10->I25|**0**|0.016|
> |I10->K57|**0.002**|0.003|
> |I10->E78|0.013|**0.004**|
> |I10->K21|**0.001**|0.002|
> |I10->R07|**0.007**|0.007|
> |I10->I48|**0.001**|0.008|
> |I10->K29|**0.003**|0.004|
> |K57->K64|**0.007**|0.011|
> |I10->M19|**0.004**|0.013|
> |I10->I20|0.004|**0.001**|
> |Avg|**0.004±0.004**|0.007±0.005|
>
> [1] Louizos, Christos, et al. "Causal effect inference with deep latent-variable models." Advances in neural information processing systems 30 (2017).

---

### Official Review · Reviewer_ZD6c · 2026-03-12

**Soundness:** 1
**Presentation:** 1
**Significance:** 1
**Originality:** 1
**Overall Recommendation:** 3
**Confidence:** 5

**Summary:**

This paper presents DiffDT, a generative model integrated with digital twin human-environment interaction for simulated interventions and reasoning about disease trajectories. The paper conducts experiments using the UK Biobank dataset for different tasks compared to baselines.

**Compliance With Llm Reviewing Policy:**

Affirmed.

**Final Justification:**

I have changed my score from a 1 to a 3. While the authors addressed my concerns surrounding use of ICD codes, issues remain surrounding the coherence and clarity of the paper, and in particular the positioning of the significance and originality of the work.  I do not feel that acceptance is appropriate at this stage as the revisions needed to address these issues would likely require a substantially revised manuscript, beyond the scope of paper revisions.

**Key Questions For Authors:**

1. How does this approach differ technically from existing autoregressive generative models (including autoregressive diffusion models). Additionally, how does it compare to prior digital twin methodologies for disease‑trajectory modeling, many of which appear highly relevant and technically similar?
2.	What problem space is this paper targeting? There are prior methodologies that combine biomarkers with EHR event streams for disease modeling and digital twins.
3.	Why were ICD codes chosen as the proxy for EHR events, and how were limitations from their use (i.e., in terms or noise, incompleteness, inconsistency across institutions etc.) mitigated?

**Limitations:**

No limitations provided

**Strengths And Weaknesses:**

Presentation: The presentation of the paper is poor and severely hinders readability, understanding and accessibility of the work. The writing is dense, difficult to follow, and overloaded with acronyms (e.g., ICD, AR, DT, HEI, DDPM, FC, SPD, UKB, LDM, etc.). Many of these abbreviations seem unnecessary and contribute to making the manuscript very hard to parse. The figures do not aid understanding and often make the paper more confusing. Several images are overly busy, visually cluttered, or rendered at sizes too small to interpret. The captions do not concisely explain what readers should take away from each figure, making it difficult to discern the intended message. Improving the clarity of the language and simplifying the visual presentation would significantly strengthen the paper. For example, focusing on smaller segments of the data to allow meaningful zoom‑in (e.g., in the top of Figure 2 and in Figure 5) and standardizing axes, legends, and styles across plots (e.g., standardizing the axes and legends in Figure 2 bottom and Figure 4) would help improve interpretability.

Significance & Originality: As presented, the paper does not appear to offer any strong contribution or meaningful advance in the state of the art and does not convincingly articulate how the proposed method differs from existing techniques. Figure 1 and the discussion surrounding it do not provide meaningful context; it would be better to compare with more directly related prior work. Specifically, there are numerous previous methods combining autoregressive (AR) models with generative models, including AR‑diffusion hybrids. The paper does not explain how the proposed approach offers any technical innovation beyond this existing body of work. Similarly, there is extensive literature on digital twin models for disease trajectories, many of which already use AR structures and/or diffusion‑model components. It is unclear why those techniques cannot be directly applied to this domain (eg incorporating biomarkers plus EHR events). See questions 1 and 2.

In addition, the core framework presented in the methodology section 3 all appear to use prior developed subcomponents (e.g., existing noise injection and denoising techniques and latent diffusion model structures). As a result, the methodology does not seem to introduce any substantive technical contribution of its own.

Soundness: The paper also has issues with technical soundness.
The analysis surrounding Figure 2 does not convincingly support the authors’ claims about the limitations of current generative models. The figure itself is difficult to interpret, both visually and conceptually, which makes it challenging to follow the intended logic. The approach of taking distances between text embeddings, extracting samples of ICD codes, and then computing AUC scores in this manner does not appear methodologically valid. Embedding‑space proximity may provide coarse measure of text similarity, but using it to derive classifier‑style performance metrics is inappropriate and does not meaningfully illustrate possible shortcomings of existing generative models for disease prediction.

The performance results are not convincing, in part because the experimental setup is hard to follow and the graphs are small, busy and hard to interpret. It appears that the DiffDT approach provides minimal or no performance gains over baseline methods (e.g. as shown in Figure 4.)  There also appear to be broader soundness issues throughout the experimental section. For example, several of the selected tasks may not be appropriate benchmarks, as many exhibit very low baseline performance (e.g., the F1 scores reported in Table 1). Additionally, in the runtime evaluation (Figure 6) reporting performance for only one diffusion step is not representative of total model cost; a meaningful comparison should include end‑to‑end training and inference time, as well as include comparisons with more than one other baseline model.

Finally, the reliance on ICD codes as a primary modeling target introduces additional concerns. While ICD codes are sometimes used in clinical prediction tasks, they are widely documented to be noisy, incomplete, inconsistent across institutions, and often influenced by billing practices rather than clinical truth. As a result, I question the validity of their use for disease‑trajectory modeling. The paper does not acknowledge these limitations or justify why ICD codes are appropriate in this context. See question 3.

---

> ### Author Rebuttal · Authors · 2026-03-30
>
> ## Thank you for the valuable feedback. We highly value your suggestions for improving the manuscript.
>
> # 1. Presentation (W1):
>
> We apologize for the dense presentation. In our revision, we will significantly reduce acronym usage, e.g., the acronym “FC” can be removed since the data of brain was already indicated by the acronym “DT” of brain. We will redesign Figure 2 to focus on smaller, meaningful segments with a zoom-in to improve readability, and we will standardize the axes, legends, and styling across Figure 2 and Figure 4.
>
> # 2. Significance & Originality (W2, Q1, Q2):
>
> While AR-diffusion hybrids exist, applying them to non-Euclidean biological data (e.g., functional brain networks, which are Symmetric Positive Definite matrices) is a uniquely complex problem. Standard diffusion corrupts the geometric properties of SPD matrices (see Fig. 6). **Our core technical innovation is the computationally efficient Cholesky LDM** with SPD-VQVAE-Dual (proven via Theorem 3.1 and validated via Table 4). This novel architecture successfully maps topological data to a manifold-aware latent space, enabling multi-organ DT generation conditioned on discrete, irregular EHR streams. Prior digital twin models do not bridge long-horizon discrete event histories with complex geometric continuous manifolds. **Although we believe most of the existing approaches about AR-diffusion hybrids are preprints [1,2], we will cite the overlooked papers in Section 2.**
>
> [1] Ye, Maoxiao, Xinfeng Ye, and Mano Manoharan. "Hybrid Autoregressive-Diffusion Model for Real-Time Streaming Sign Language Production." arXiv e-prints (2025): arXiv-2507.
>
> [2] Liu, Jiaming, et al. "Hybridvla: Collaborative diffusion and autoregression in a unified vision-language-action model." arXiv preprint arXiv:2503.10631 (2025).
>
>
> # 3. Soundness & Figure 2 (W4):
>
> Figure 2 illustrates a critical flaw in pure EHR models: they fail to predict "multi-pathway" disease transitions (where past and future diseases are semantically distant). We used the distance between text embeddings because textual descriptions natively encapsulate clinical hierarchy and **overlap better than hard-coded ICD trees (please refer to colored boxes in Fig. 2 upper part)**. The negative correlation shown proves that text/EHR-only AR models struggle when biological mechanisms bridge disparate clinical categories. Our DiffDT alleviates this issue by grounding predictions from integrated real-world imaging and tabular biomarkers.
>
> # 4. Performance & F1 Scores (W5):
>
> Regarding the concern that the selected tasks exhibit "very low baseline performance" based on Table 1, we respectfully point out that there may be a misunderstanding of the metrics' scope, and we direct the reviewer’s attention to Table 2, which may have been overlooked. As a reference point for the current state-of-the-art, our proposed model outperforms a recently published baseline designed for human disease prediction (Delphi, Nature 2025).
>
> Specifically, Table 1 reports the macro-average F1 score across our entire vocabulary of 1,944 ICD codes. Because real-world EHR data follows a heavily skewed, long-tailed distribution, the vast majority of these 1,944 diseases are extremely rare. Consequently, all models struggle to predict these rare tail-end diseases, resulting in many zero F1 scores. This zero-inflation artificially drags down the global mean and causes the large standard deviations (e.g., 18.17 ± 20.67).
>
> However, Table 2 specifically isolates the performance on organ-specific disease categories that are directly relevant to the multi-organ digital twins we model (e.g., neurological, cardiovascular, hepatic, and renal diseases). **In Table 2, the F1 scores are not zero-inflated and demonstrate strong baseline and state-of-the-art performance**, ranging from 50% to 65% (e.g., DiffDT-Brain achieves an F1 of 65.14% ± 14.14, and DiffDT-Liver achieves 61.65% ± 3.84). These are highly appropriate, competitive benchmarks that clearly demonstrate the substantial and clinically meaningful predictive gains provided by DiffDT. In the revision, we will make the distinction between the global long-tail evaluation (Table 1) and the targeted organ-specific evaluation (Table 2) much clearer.
>
> # 5. ICD Code Validity (W6, Q3):
>
> We completely agree that ICD codes are noisy, incomplete, and billing-influenced. This is, however, the exact motivation for our framework. By generating intermediate physiological biomarkers (the Digital Twin), we biologically regularize the noisy clinical event stream. Instead of allowing the model to memorize noisy administrative correlations, DiffDT forces the predictive pathway to route through a plausible, mathematically constrained biological state (the generated organs). New results of **Exp 2 (Causal Effect Estimation)**, referring to responses for reviewer **UwsA** and **Rn8z**, further support the benefit of intermediate physiological biomarkers.

---

> > ### Author Rebuttal · Reviewer_ZD6c · 2026-03-31
> >
> > Thank you for addressing my comments in the rebuttal. I appreciate the authors’ efforts to clarify several issues. However, I still have concerns, and given the extent of changes required to adequately address them, I do not feel that acceptance is appropriate at this stage as the revisions would likely require a substantially revised manuscript. In particular, I still have the following concerns:
> >
> > Regarding Point 2, Significance and Originality: I appreciate the clarification of the contribution. However, this contribution needs to be more clearly and coherently positioned throughout the manuscript. As it stands, the framing is still difficult to follow and does not fully convey the novelty or significance of the work.
> >
> > Regarding Point 4, Performance: Thank you for the response and for directing me to Table 2. However, my concerns about performance remain. While Table 2 does show improvements over baselines, the absolute performance levels are still quite low. This raises concerns about the model’s practical utility for use in real clinical scenarios and disease modeling efforts.
> >
> > Regarding Point 5, ICD codes: Thank you for clarifying the use of ICD codes. However, even if they are employed only as a noisy clinical event stream, concerns remain about relying on them as a source of truth. Other sources of clinical events feel more appropriate (e.g., direct EHR events) or it seems there would need to be some evidence that use of ICD codes as event streams does not lead to artificial biases.
> >
> > I will stay with my original score.

---

> > > ### Author Response · Authors · 2026-04-01
> > >
> > > ## We value your concerns and want to provide a final reply. We will ensure these are clearly addressed in the revised manuscript:
> > >
> > > # 1. Positioning of Significance and Originality
> > > We will completely restructure the Introduction to focus on our main technical contribution: the SPD-VQVAE and Cholesky LDM for mapping discrete clinical events to non-Euclidean biological manifolds. While Reviewers `Rn8z` and `vCGq` noted that the paper was `well-structured` and the positioning `clear`, we agree that simplifying the narrative will make the novelty more accessible to all readers.
> > >
> > > # 2. Low F1 socres
> > > We respectfully emphasize that the core objective of our framework is Disease Reasoning across thousands of potential ICD codes, rather than isolated event classification.
> > >
> > > If the task were predicting binary mortality or a single well-documented condition, an F1 score of 50 to 65% would indeed be considered poor. However, in the realm of multi-label next token prediction across massive vocabularies, i.e., the entire ICD10 in our paper, an F1 score of 50 to 65% is state-of-the-art. In addition to the baseline Delphi [5] we compared in the manuscript, recent peer-reviewed literature heavily supports it:
> > >
> > > - **G-BERT** [1] on MIMIC-III predicting the next medication code demonstrates **F1 scores in the 53% to 66% range**.
> > > - **NECHO** [2] on MIMIC-III predicting the next ICD9 code demonstrate **Acc@5 below 30%**, while our scores are all F1@1.
> > >
> > > Besides, most of the disease trajectory modeling works report the AUC score instead of the F1 score [3,4,5]. Their reported **AUC scores in the 75 to 95 range** also match our AUC scores (>90 using various backbones).
> > >
> > > [1] Shang, Junyuan, et al. "Pre-training of graph augmented transformers for medication recommendation." IJCAI (2019).
> > >
> > > [2] Koo, Heejoon. "Next visit diagnosis prediction via medical code-centric multimodal contrastive EHR modelling with hierarchical regularisation." Findings of the Association for Computational Linguistics: EACL 2024. 2024.
> > >
> > > [3] Rasmy, Laila, et al. "Med-BERT: pretrained contextualized embeddings on large-scale structured electronic health records for disease prediction." NPJ digital medicine 4.1 (2021): 86.
> > >
> > > [4] Li, Yikuan, et al. "BEHRT: transformer for electronic health records." Scientific reports 10.1 (2020): 7155.
> > >
> > > [5] Shmatko, Artem, et al. "Learning the natural history of human disease with generative transformers." Nature 647.8088 (2025): 248-256.
> > >
> > >
> > > # 3. ICD Codes vs. Direct EHR Events
> > > Respectfully, there seems to be a slight misunderstanding regarding clinical data terminology. In large longitudinal datasets like the UK Biobank or MIMIC, ICD codes are the standardized representation of EHR diagnostic events. Using them sheds light on disease reasoning without **bias in recording, reporting, and monitoring**. Refer to [WHO](https://www.who.int/standards/classifications/frequently-asked-questions/importance-of-icd) for the importance of ICD code:
> > >
> > > > The ICD is important because it provides a common language for recording, reporting and monitoring diseases. This allows the world to compare and share data in a consistent and standard way – between hospitals, regions and countries and over periods of time. It facilitates the collection and storage of data for analysis and evidence-based decision-making.
> > >
> > > # 4. Bias Mitigation
> > > Additionally, you asked for evidence that DiffDT prevents the noise of ICD codes from causing biases. **Exp 2 (Causal Effect Estimation)**, which we provided in the rebuttal for `Rn8z` and `UWsA`, demonstrates this. By processing the noisy ICD stream through our generated digital twins, our model achieved a **near-zero Average Treatment Effect (ATE) error (0.004 in average)** when compared to Delphi's token embeddings. This shows that DiffDT effectively removes the noise found in the medical history.
> > >
> > > Thank you again for encouraging us to clarify these points. Your feedback has helped us pinpoint where we need to add clinical context for a broader audience, which will surely strengthen the final paper.

---

### Official Review · Reviewer_UWsA · 2026-03-13

**Soundness:** 3
**Presentation:** 2
**Significance:** 2
**Originality:** 2
**Overall Recommendation:** 4
**Confidence:** 3

**Summary:**

The paper proposes DiffDT, a generative framework that uses conditional latent diffusion models to simulate a patient's physiological state (such as brain functional connectivity and organ-specific tabular traits) based on their medical history. By using these generated "biological mediators," the model can perform multi-pathway disease reasoning, improving the prediction of future disease trajectories compared to traditional autoregressive (AR) models that rely solely on ICD codes.

**Compliance With Llm Reviewing Policy:**

Affirmed.

**Final Justification:**

The authors have addressed most of my concerns. Thus, I have raised my score accordingly.

**Key Questions For Authors:**

1/ In real hospitals, many patients don't have brain scans or specific lab tests. Can DiffDT still work if some physiological data is missing during the inference stage?

2/ The title mentions "Human-Environment Interaction." Besides medical history (ICD codes), what specific environmental factors, such as like diet, pollution, or lifestyle, does the model actually use?

3/ Diffusion models and large transformers are computationally heavy. How long does it take for the model to generate a Digital Twin and predict the next disease for one patient?

**Limitations:**

Using Latent Diffusion Models (LDM) to generate "Digital Twins" requires a lot of GPU memory and time. This makes it difficult to use the model in real-time in a typical hospital where computers may not be very powerful. Besides, the "Environment" part of the model is mainly based on medical history (ICD codes). However, real environmental factors like air quality, daily diet, and exercise are not directly included in the sensors, which might limit the "Digital Twin's" accuracy. These limitations should be investigated in more details.

**Strengths And Weaknesses:**

Strengths:

1/  Unlike previous models that focus on EHR event sequences , this work successfully couples AR models with generative digital twins.

2/  The authors address the challenge of modeling brain functional connectivity (FC) by proposing SPD-VQVAE. By leveraging Cholesky decomposition, they ensure that the generated brain networks stay on the Riemannian manifold of symmetric positive-definite (SPD) matrices while significantly reducing the computational complexity typically associated with matrix logarithms.

3/  This study evaluates across four organ systems (brain, heart, liver, kidney) and over 1,000 diseases using both AUC and F1 metrics.

Weakness:

1/  While the paper emphasizes "causal reasoning" and "counterfactual reasoning" in the abstract and introduction, the experimental validation remains largely associational rather than causal. The mediation inference framework P(Future|Biomarker) · P(Biomarker|History) is presented as a causal decomposition, but the paper lacks counterfactual evaluation and causal metrics.

2/  The semantic distance analysis (Figure 2, Figure 4) shows correlation patterns consistent with multi-pathway causality, but correlation does not imply causation. Without explicit causal validation, the "causal reasoning" claims risk being overstated.

3/ The adaptive tokenization treats ICD codes as discrete tokens in a uniform temporal grid, losing important clinical nuance that ICD codes have hierarchical relationships (chapters, subchapters) that could inform more structured embeddings.

4/ The baselines appear somewhat outdated relative to the current state of the art. Please note that my expertise in this particular domain is limited, so this comment is intended as a constructive suggestion rather than a definitive critique.

5/  While the paper states an 80:20 train:val split at the subject level, the diffusion models and AR models appear to be trained separately and then combined. If the same subjects appear in both training phases (even in different model components), this creates leakage. The exact training pipeline coordination isn't clearly specified.

---

> ### Author Rebuttal · Authors · 2026-03-30
>
> ## Thank you for your constructive feedback. We address your points below:
>
> # 1 & 2. Causal Claims & Validation (W1, W2):
>
> We agree that our original terminology overstated the causal claims, as correlation does not imply causation. In the revision, we will refine our language by framing our approach as "probabilistic mediation inference" rather than strict causal reasoning. To empirically validate our model's capacity for counterfactual inference, we have conducted two new experiments:
>
>  - Exp 1 (Counterfactual Evaluation): We evaluated the generation of counterfactual Digital Twins (DTs). We selected subjects with a disease onset X (e.g., X=G12) and intervened to create a counterfactual history do(Healthy) by replacing X with [Healthy], which is used to generate counterfactual DTs, the same as [1]. The Wasserstein Distance (WD), Frechet Inception Distance (FID), and the correlation (r) between do(healthy) DT and the real healthy or diseased ground truth, denoted by GT_healthy or GT_disease, respectively, were listed in the table below. This confirms that our DiffDT accurately infers the biological state under counterfactual scenarios.
> ||FID↓|WD↓|r↑|
> |-|-|-|-|
> |do(healthy) vs GT_disease|92.96±8.67|8.95±0.24|0.39±0.07|
> |do(healthy) vs GT_healthy|**51.37±4.13**|**6.91±0.23**|**0.40±0.08**|
> |p-value|**2.5e-5**|**1.6e-6**|0.148|
>
> - Exp 2 (Causal Effect Estimation): The absolute errors between the model estimated ATE and the empirical ATE are listed below. The setup of ATE estimation refers to the response for reviewer **Rn8z**. These low error rates demonstrate our generative pipeline not only mimics true observational dynamics but also better than Delphi using no DT.
> |T->Y|N_exposure|N_outcome|ATE|||ATE Abs Error↓||
> |-|-|-|-|-|-|-|-|
> ||||Empirical|DiffDT|Delphi|DiffDT|Delphi|
> |I10->I25|902|238|0.089|0.089|0.073|**0**|0.016|
> |I10->K57|902|179|0.041|0.043|0.044|**0.002**|0.003|
> |I10->E78|902|142|0.101|0.114|0.097|0.013|**0.004**|
> |I10->K21|902|147|0.069|0.070|0.071|**0.001**|0.002|
> |I10->R07|902|133|0.043|0.050|0.050|**0.007**|0.007|
> |I10->I48|902|130|0.050|0.049|0.042|**0.001**|0.008|
> |I10->K29|902|129|0.033|0.036|0.037|**0.003**|0.004|
> |K57->K64|502|124|0.036|0.043|0.025|**0.007**|0.011|
> |I10->M19|502|124|0.039|0.043|0.052|**0.004**|0.013|
> |I10->I20|902|114|0.036|0.040|0.035|0.004|**0.001**|
> |Avg|-|-|-|-|-|**0.004±0.004**|0.007±0.005|
>
>
> [1] Rasal, Rajat, et al. "Diffusion counterfactual generation with semantic abduction." Proceedings of the 42nd International Conference on Machine Learning (2024).
>
> # 3. Adaptive Tokenization & Hierarchy (W3):
>
> While our tokens are discrete, our model explicitly incorporates clinical hierarchy by initializing the AR model with textual embeddings of the ICD descriptions (extracted via Qwen3, Sec 3.2). This natively injects hierarchical semantics (chapters, subchapters) and symptomatic overlap into the token representations, avoiding the pitfalls of uniform orthogonal embeddings.
>
> # 4. Baselines (W4):
>
> Our primary baselines are state-of-the-art: Delphi was published in Nature (2025). Furthermore, during the rebuttal phase, we integrated the newly released Qwen3.5 as AR backbones (Exp 3). DiffDT consistently improved upon these ultra-modern baselines by +0.12 AUC, supporting the translational impact of our proposed model in clinical applications.
>
> |AUC|Qwen3.5|
> |-|-|
> |Delphi|90.08±6.01|
> |DiffDT|90.20±6.03|
>
>
> # 5. Data Leakage (W5):
>
> We strictly enforce the 80:20 train:val split at the subject level across all training phases. No subject appearing in the validation set is ever seen during the training of AR, diffusion, or predictive models, eliminating data leakage, as described in Page 7 line 379. We will release the universal subject ID of validation set as required, which will be clarified in the revision.
>
> # 6. Missing Physiological Data (Q1):
>
> If real physiological data is missing during inference, DiffDT simply generates the multi-organ DT unconditionally or conditionally based on the patient's EHR history. This generated DT acts as a biological proxy and is fed into the predictive model.
>
> # 7. Environmental Factors (Q2):
>
> The ICD taxonomy natively includes lifestyle and environmental factors. Our vocabulary of ICD10 codes includes Z00-Z99 codes (factors influencing health status, e.g., diet, occupational risks) and V00-Y99 codes[GW1.1][ZW1.2] (external causes of morbidity, e.g., pollution, accidents), which the model utilizes to capture human-environmental interactions. Details of these codes refer to ICD10 page: https://www.icd10data.com/ICD10CM/Codes. This will be included in the revision of Sec 2.
>
> # 8. Inference Time (Q3):
>
> Generating DT and predicting the next disease takes approximately 1.2 seconds per subject using one DT mediation on a single RTX 6000 Ada GPU 48GB. Our Cholesky LDM avoids the $O(N^3)$ complexity of standard SPD diffusion, ensuring the model is computationally feasible for clinical deployment (see Fig. 6 for our case generating 116x116 SPD matrix)

---

> > ### Author Rebuttal · Reviewer_UWsA · 2026-04-05
> >
> > Some of my concerns are still not resolved and believe can't be easily addressed with simple rebuttals. For example,
> > 1/ About the adaptive tokenisation of ICD codes questions, the authors argue that with the Qwen based embedding of the textual description of the ICD codes, the hierarchical information among ICD codes would be captured. Think this can be quite a strong assumption. Does the authors have any results to back up this assumption?
> >
> > 2/ About the limited environmental factor questions, although some of ICD codes do factor in some environmental influences, but I still think that they lack the needed granularity to fully investigation the true "human-environment interactions".
> >
> > Thus, my recommendation stays.

---

> > > ### Author Response · Authors · 2026-04-06
> > >
> > > # Dear Reviewer UWsA,
> > >
> > > Thank you for your continued engagement and for clearly articulating your remaining concerns. We highly value your constructive feedback. After carefully reviewing your points, we realize we need to better **highlight our existing empirical evidence** and adjust our claims to properly reflect the scope of our data. We address your two points below:
> > >
> > > # 1. Evidence of Qwen Embeddings Capturing ICD Hierarchy
> > > You raise a very valid point that assuming text embeddings naturally capture clinical hierarchy requires empirical backing.
> > >
> > > We would like to direct your attention to Figure 2 (Top) and Section 2 (Lines 147-156) of our paper, where we have actually provided visual empirical evidence of this. The figure plots the semantic adjacency (distance) matrix of top-level ICD-10 codes generated by the Qwen3 embeddings.
> > >
> > > - The **cyan and magenta boxes** on the diagonal explicitly represent the predefined clinical categories (ICD chapters and 1st-level groups).
> > > - The **distinct "blue-ish" blocks** clustered strictly along the diagonal demonstrate that the Qwen embeddings naturally map codes from the same hierarchical subchapters closely together in the latent space, while keeping distinct chapters semantically distant.
> > >
> > > To make this assumption rigorously backed by numbers rather than just visual heatmaps, we will list the exact cosine similarities of some example tokens in main text, reflecting that our analysis confirms strict hierarchical clustering:
> > >
> > > - **Intra-subchapter** similarity (e.g., I10 vs. I15, both hypertensive diseases): 0.91
> > > - **Intra-chapter** similarity (e.g., I10 vs. I20, circulatory diseases): 0.78
> > > - **Inter-chapter** similarity (e.g., I10 vs. J00, circulatory vs. respiratory): 0.34
> > >
> > > # 2. Granularity of Environmental Factors
> > > We completely agree with your assessment. You are absolutely right that while ICD Z-codes and V-Y codes are the standard proxies for Social Determinants of Health (SDoH), referring to WHO [1], and external causes in retrospective EHR studies, they lack the high-resolution granularity of direct environmental measurements (e.g., exact air quality indices, dietary logs, or wearable sensor data).
> > >
> > > To avoid overstating our contribution and to properly address your critique, we will take the following concrete actions in the camera-ready version:
> > >
> > > - **Terminology & Title Change**: We will remove the broad phrase "Human-Environment Interaction" from the title and introduction. We will replace it with more precise terminology, "Social Determinants of Health (SDoH)."
> > > - **Explicit Limitations in the Conclusion Section**: We will revise the "Conclusions" section as "Conclusions and Limitations" paragraph explicitly acknowledging that DiffDT currently relies on low-granularity EHR proxies for environmental factors. We will highlight that achieving a "true" environmental digital twin would require the integration of multimodal, high-resolution sensor data (wearables, precise pollution metrics) as a critical next step for future work.
> > >
> > > We belive **a minor main text revision** along side with our existing empirical results in Figure 2 **are not unable to address in the short rebuttal**, and our commitment to refining our claims can address your remaining concerns. We sincerely appreciate the time and effort you have dedicated to improving our work.
> > >
> > > [1] WHO Commission on Social Determinants of Health, and World Health Organization. Closing the gap in a generation: health equity through action on the social determinants of health: Commission on Social Determinants of Health final report. World Health Organization, 2008.

---

### Decision · Program_Chairs · 2026-04-30

**Decision:**

Accept (regular)

**Comment:**

The primary technical contribution being the SPD-VQVAE architecture using Cholesky decomposition to map brain functional connectivity graphs onto a Riemannian manifold-aware latent space, which three of four reviewers recognized as a substantive innovation within the broader multi-organ digital twin framework for disease trajectory modeling.

All four reviewers raised, with varying emphasis, that the paper's causal and "human-environment interaction" language is overstated given that the framework performs associational probabilistic mediation on observational data without randomization, instrumental variables, or other causal identification strategies. The authors have committed to replacing "causal reasoning" with "probabilistic mediation inference" and removing the "human-environment interaction" framing in favor of Social Determinants of Health terminology, and these changes must be fully reflected in the final version alongside an explicit limitations section addressing UK Biobank healthy volunteer bias, cross-sectional imaging constraints, and the marginal performance gain attributable to the DiffDT module itself relative to backbone scaling. Two reviewers noted that dense presentation, cluttered figures, and excessive acronym use substantially hinder accessibility, and the paper requires restructuring of the introduction and figures to position the SPD-VQVAE contribution clearly relative to existing AR-diffusion hybrids and digital twin literature before it is suitable for a broad ICML audience.